# Exploring the toxicological mechanisms of Benzo[a]anthracene (BaA) exposure in lung adenocarcinoma (LUAD) via network toxicology, machine learning, and multi-dimensional bioinformatics analysis

Zhiyao Shi[1,2], Zhiyong Fang[2], Qiang Qin[1], Yu Gao[2], Xi Yang[2], Likun Liu[2], Xixing Wang[2]*

1 The First Clinical College, Shanxi University of Traditional Chinese Medicine, Taiyuan, Shanxi, China,
2 Department of Oncology, Shanxi Provincial Hospital of Traditional Chinese Medicine, Taiyuan, Shanxi, China

* wangxx315@163.com

## Abstract

### Background

Lung adenocarcinoma (LUAD) is a major lung cancer subtype influenced by environmental factors. Benzo[a]anthracene (BaA), a common Group 2B carcinogen found in pollutants, smoke, and food, shows genotoxic and oncogenic activity; however, its specific mechanisms in LUAD pathogenesis remain unclear and warrant systematic investigation.

### Objective

This study aims to elucidate the mechanisms of BaA-induced LUAD, identify core targets, validate their expression, immunorelevance and clinical significance, and construct a hypothesis framework for AOP in BaA-exposed LUAD.

### Methods

We integrated network toxicology, multi-machine learning algorithms (LASSO, SVM-RFE, and Random Forest) and multidimensional bioinformatics analysis. Potential BaA-LUAD intersection targets were collected from public databases and subjected to functional enrichment analysis. Core targets were screened and validated using GEO and TCGA-LUAD (via UALCAN) datasets for differential expression, immune infiltration and prognostic value. Molecular docking and 100 ns molecular dynamics (MD) simulations were applied to evaluate the binding stability between BaA and core targets.

**Data availability statement:** All relevant data and analysis scripts are available from the Zenodo repository: https://doi.org/10.5281/zenodo.17747122. This includes the R/Python scripts for statistical analysis and visualization, as well as the parameter files for molecular docking and dynamics simulations.

**Funding:** This work was supported by: State Administration of Traditional Chinese Medicine of the People's Republic of China (2022-245, 202203, 2018-131) to Xixing Wang; Department of Science and Technology of Shanxi Province (02103021224437, [2019]61) to Xixing Wang; Health Commission of Shanxi Province (2020TD04, 2022XM10) to Xixing Wang. The funders had no role in study design, data collection and analysis, decision to publish, or preparation of the manuscript.

**Competing interests:** The authors have declared that no competing interests exist.

## Results

A total of 248 intersection targets were identified, with significant enrichment in chemokine signaling, ErbB signaling, and viral protein–cytokine receptor interaction pathways. Machine learning prioritized five core targets: *TNNC1, ABCC3, CRABP2, CXCL12,* and *OLR1*. These genes were consistently dysregulated in LUAD samples across cohorts ($p < 0.05$) and correlated distinctly with immune cell infiltration: *TNNC1* was associated with anti-tumor immunity, while the others linked to immunosuppressive cells. Prognostic analysis showed trends of *ABCC3/CRABP2* high-expression and *TNNC1/CXCL12/OLR1* low-expression correlating with patient outcomes ($p > 0.05$). Molecular docking confirmed stable binding between BaA and all core targets, with the strongest affinity for CRABP2 (−8.4 kcal/mol). MD simulations further supported complex stability.

## Conclusion

BaA promotes LUAD progression via multi-target regulation and tumor immune microenvironment remodeling. This study offers an integrated computational framework and an AOP-based theoretical foundation for assessing pollutant health risks and informing targeted LUAD interventions.

## 1. Introduction

Lung adenocarcinoma (LUAD) represents approximately 40% of all lung cancer cases and is a leading cause of cancer-related mortality worldwide [1]. The disease often progresses insidiously, resulting in late-stage diagnosis for most patients. Current treatments demonstrate limited efficacy against advanced LUAD, with a five-year survival rate of only 15–20% [2]. A major challenge in LUAD management stems from poorly understood pathogenic mechanisms driven by environmental carcinogens. Among these, polycyclic aromatic hydrocarbons (PAHs) are widely recognized as significant environmental risk factors. PAHs can induce oxidative stress through multiple pathways, and their phototoxicity under light exposure further promotes the generation of reactive oxygen species (ROS) such as singlet oxygen ($^1O_2$) and superoxide anion ($O_2^-$), causing severe damage to biomolecules including DNA and cellular membranes [3].

Benzo[a]anthracene (BaA), a representative PAH, is commonly generated through incomplete combustion of organic materials in vehicle exhaust, cigarette smoke, industrial emissions, and grilled foods [4]. Human exposure to BaA is widespread, with biomonitoring studies detecting BaA and its metabolites in a majority of the general population [5]. Previous research has established that BaA exerts genotoxic effects through DNA adduct formation and promotes malignant cell transformation by activating oncogenic pathways such as AGE-RAGE [6]. Although BaA alone displays weak ROS-inducing activity, it can synergistically enhance oxidative damage in PAH mixtures by targeting specific proteins including SLC1A5 to inhibit glutathione

synthesis [7]. Furthermore, similar to other PAHs, BaA undergoes metabolic activation via cytochrome P450 enzymes such as CYP1A1 and CYP1B1, producing reactive metabolites that amplify ROS production and promote tumorigenesis [8]. However, existing studies have primarily focused on general carcinogenicity, lacking systematic exploration of LUAD-specific molecular targets and regulatory networks. Moreover, no standardized adverse outcome pathway (AOP) has been established for the BaA exposure-LUAD progression axis, hindering environmental risk assessment and targeted intervention strategies.

This study employs an integrated approach combining network toxicology, machine learning, multi-dimensional bioinformatics, and the AOP framework to systematically investigate BaA-induced toxicity and molecular mechanisms in LUAD. Network toxicology, which integrates network pharmacology and systems biology to construct compound-toxicity-target networks, provides a powerful tool for deciphering pollutant-induced disease mechanisms, including those of PAHs. Machine learning algorithms were applied to identify high-confidence core toxic targets of BaA in LUAD, while multi-dimensional bioinformatics analyses elucidated their pathogenic roles in disease progression. Molecular docking and dynamics simulations further predicted interactions between BaA and its potential targets. The AOP framework was used to visualize the complete exposure-effect sequence, clarifying the chain of events from molecular initiation to adverse outcomes. By integrating BaA's toxic targets with LUAD pathogenic genes, this work provides a theoretical foundation for understanding BaA's toxicological mechanisms and supports environmental risk assessment and disease prevention strategies.

## 2. Materials and methods

### 2.1. Acquisition of BaA-related targets

The chemical structure and Simplified Molecular Input Line Entry System (SMILES) of BaA (CID: 5954) were retrieved from the PubChem database (https://pubchem.ncbi.nlm.nih.gov/) for subsequent target prediction and molecular docking [9]. Potential human (Homo sapiens) BaA targets were screened from the ChEMBL database (https://www.ebi.ac.uk/chembl/) [10], Swiss Target Prediction database (http://swisstargetprediction.ch/) [11], and Comparative Toxicogenomics Database (http://ctdbase.org/) [12]. To exclude pan-genes and select high-confidence targets, further filtering was conducted: Swiss Target Prediction targets with Probability > 0, CTD targets with Reference Count > 1. Targets were merged, and gene names were standardized using the UniProt database (https://www.uniprot.org/) [13]. Duplicates were removed to construct a BaA target library.

### 2.2. Acquisition of LUAD-related targets

In terms of disease-related genes, clinically validated LUAD-related genes were retrieved from OMIM (https://omim.org/) [14]; LUAD-related genes from GeneCards (https://www.genecards.org/) with "Relevance Score ≥ 10" [15]; and LUAD-related "Validated Targets" from TTD (https://db.idrblab.org/ttd/) [16]. Genes from the three databases were merged, duplicates removed, and standardized to form the LUAD target library.

### 2.3. Screening of intersection targets and functional enrichment analysis

Intersection targets between BaA and LUAD were identified using Venny 2.1 (https://bioinfogp.cnb.csic.es/tools/venny/) and visualized via a Venn diagram. Functional enrichment analysis for Gene Ontology (GO) terms, as well as Kyoto Encyclopedia of Genes and Genomes (KEGG) pathways, was conducted using the DAVID platform (https://david.ncifcrf.gov/) [17]. Terms with an adjusted p-value, $p < 0.05$ were considered statistically significant.

### 2.4. Screening of core targets by machine learning

Next, we applied three machine learning algorithms to identify core targets among the intersection targets between BaA and LUAD derived from 2.3.: Least Absolute Shrinkage and Selection Operator (LASSO) regression [18], support Vector

Machine-Recursive Feature Elimination (SVM-RFE) [19] and Random Forest to identify core target genes [20]. Specifical-ly,LASSO regression, implemented using the R package "glmnet", applied L1-penalization to shrink coefficients of non-informative variables to zero (parameters: standardization = TRUE, alpha = 1, family = "binomial", 3-fold cross-validation). The optimal feature subset was determined at the lambda value yielding minimum cross-validation error. SVM-RFE was performed using the "mlbench" and "caret" packages on normalized expression data, iteratively eliminating features with the lowest contribution to classification. Random Forest utilized the "randomForest" package with 500 decision trees, where feature importance was measured by Gini impurity decrease, and variables with importance scores above 10 were retained. Final core targets were the intersection of both results (Venn diagram), ensuring reliability. The core target was the intersection gene of the three machine learning methods.

### 2.5. Validation of core targets expression and diagnostic efficacy in GEO database

To validate core targets, three GEO datasets were downloaded (https://www.ncbi.nlm.nih.gov/geo/) [21]: three GEO data-sets were downloaded: GSE10072 (58 LUAD samples + 49 Normal samples), GSE32863 (58 LUAD samples + 58 Normal samples), and GSE31210 (226 LUAD samples + 20 Normal samples) Genes with adjusted p-value, $p < 0.05$ were consid-ered significantly and differentially expressed genes. After removing batch effects and standardization, probe IDs were converted to gene symbols, the three datasets were integrated, and samples were grouped by sample type. Differentially expressed genes (DEGs) were screened via limma (log2|FC| > 1, adjusted $p < 0.05$) and visualized with volcano plots/ heatmaps. Receiver operating characteristic (ROC) curves and calculated the area under the curve (AUC) values were calculated to evaluate the diagnostic potential of core biomarkers for LUAD.

### 2.6. Immune infiltration analysis of core targets

The single-sample gene set enrichment analysis (ssGSEA) was performed using the GSVA (v1.46.0) R package to exam-ine immune cell infiltration scores between Normal and LUAD samples in the GEO datasets (GSE10072, GSE32863, and GSE31210) [22]. This analysis focuses on examining the correlation between core target expression and immune cell infiltration levels, aiming to investigate the potential regulatory role of core targets in the immune microenvironment of LUAD [23].

### 2.7. Validation of core targets using the UALCAN database

To further confirm the clinical relevance and prognostic significance of the identified core targets in LUAD, we utilized the UALCAN platform (http://ualcan.path.uab.edu) [24], which integrates TCGA multi-omics and clinical data [25]. Using UAL-CAN's "Expression Analysis – LUAD" module, we compared the mRNA expression levels of the core targets between 515 LUAD samples and 59 Normal samples. Statistical significance was assessed via Student's t-test, and results were visu-alized using box plots. Prognostic value was evaluated with the "Survival Analysis" module, where patients were stratified into high-expression and low-expression groups based on median expression levels. Kaplan–Meier survival curves were generated, and differences in overall survival were assessed using the log-rank test.

### 2.8. Molecular docking

To investigate the molecular interactions between BaA and the core targets, a semi-flexible molecular docking approach was employed. The two-dimensional structure of BaA was obtained from the PubChem database and converted into its three-dimensional conformation using Open Babel software, with the output saved in.mol2 format. The three-dimensional structures of the five core targets were sourced from the RCSB PDB (https://www.rcsb.org/) [26], ensuring the use of accurate, experimentally validated protein structures. Prior to docking, protein structures were prepared using PyMOL 3.2 by removing any pre-existing ligands and water molecules, and the resulting structures were saved as.pdb files. Subsequent processing was performed with AutoDock Tools 1.5.7, where proteins were dehydrated and hydrogenated

to optimize the structures for docking simulations. These steps ensured the proteins were in a proper state for interaction with BaA. A grid box was defined for each target protein to specify the active site (maximum presumed binding cavity identified by AutoDock Tools through computational detection), ensuring the docking process was focused on biologically relevant binding. Each docking experiment was independently repeated three times to verify the consistency of the results. The Vina binding affinity (kcal/mol) served as the primary metric for evaluating binding affinity, and the conformation with the lowest binding affinity was selected as the optimal docking result. The stability of the resulting conformations was further assessed by verifying hydrogen bonding and hydrophobic interactions. Finally, visualization was carried out using PyMOL 3.2 [27].

### 2.9. Molecular dynamics simulation

The molecular docking results were screened to identify complexes with optimal binding energies for molecular dynamics (MD) simulations. The procedure involved: 1) Using PyMOL to dehydrate and hydrogenate the docking output files, then saving them as protein receptor files (PDB format) and small molecule ligand files (MOL2 format); 2) Generating protein topology using Amber and small molecule topology parameters with SObTop; 3) Merging the topology files, followed by box addition, water model insertion, ion incorporation, charge equilibration, energy minimization (steepest descent/conjugate gradient), conformational constraints, NVT pre-equilibration, and NPT pre-equilibration. After formal simulations, the results underwent trajectory correction, translational and rotational removal, and calculation of key metrics (RMSD, RMSF, RG, hbond_num, SASA, resarea, atomarea, dgsolv, volume). Finally, the results were visualized using DulvyTools software.

### 2.10. The construction of AOP hypothesis

Phenotypes closely associated with BaA exposure and LUAD progression were selected as potential key events (KEs). Based on target–phenotype relationships and gene–phenotype networks, genes linked to these phenotypes were defined as candidate molecular initiating events (MIEs). By integrating evidence levels and biological plausibility reported in the literature, we constructed a hypothesized AOP to conceptually link BaA exposure with LUAD development.

### 2.11. Ethics statement

This study is a bioinformatics analysis based on publicly available datasets and does not involve direct experimentation on humans, animals, or human/animal-derived biological samples. No ethical approval or informed consent was required for this study, as all data used are de-identified and publicly accessible in compliance with relevant database policies.

## 3. Results

### 3.1. Network toxicology analysis of potential targets for BaA-induced LUAD

As an advanced analytical approach, network toxicology predicts environmental pollutants' mechanisms and targets. To explore BaA's potential pathogenic effects on LUAD, this study adopted it. BaA targets: 432 from ChEMBL, 4 from Swiss Target Prediction, 12 from CTD; 444 unique after deduplication (Fig 1A, S1 Table). LUAD targets: 2392 from GeneCards, 5120 from OMIM, 8 from TTD; 6461 unique after deduplication (Fig 1B, S2 Table). Their intersection yielded 248 targets, regarded as potential BaA-induced LUAD targets (Fig 1C, S3 Table). These results indicate that BaA may induce LUAD through regulating a set of common targets overlapping with LUAD pathogenic genes.

### 3.2. Functional enrichment analysis of BaA-induced LUAD-related intersection targets

To explore the mechanism by which BaA induces LUAD, we conducted an enrichment analysis of 248 overlapping target genes. We selected the top 10 GO terms with the lowest false discovery rate (FDR) in each category and visualized them

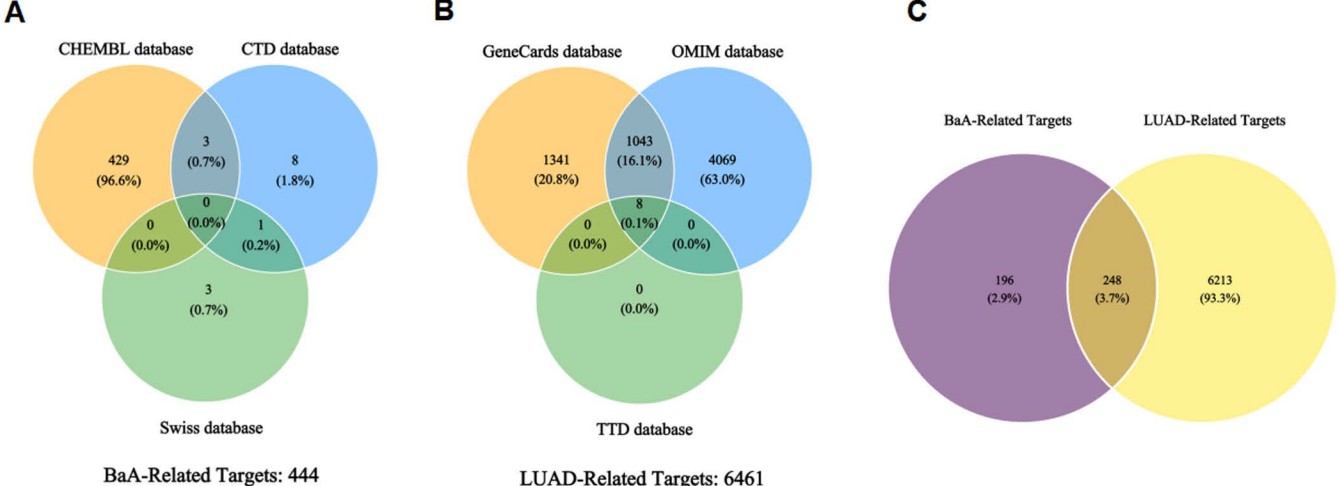

**Fig 1. Venn diagram showing the overlap of BaA-related targets and LUAD-related targets. (A)** Overlap of BaA-related targets identified from ChEMBL, Swiss Target Prediction, and CTD databases. **(B)** Overlap of LUAD-related targets identified from GeneCards, OMIM, and TTD databases. **(C)** Overlap of intersection targets for BaA-induced LUAD.

(Fig 2A). In the BP category, targets were primarily enriched in "chemokine−mediated signaling pathway", "response to chemokine" and "cellular response to chemokine", implying roles in chemokine-related signaling and cell chemotaxis; in the CC category, top enriched terms were "external side of plasma membrane", "membrane raft" and "membrane microdomain", suggesting localization in membrane structures and vesicular compartments to support signal transduction and secretion; while in the MF component, significant enrichment was found in "C−C chemokine receptor activity", "C−C chemokine binding" and "G protein−coupled chemoattractant receptor activity", indicating core functions in chemokine binding and G protein−coupled receptor signaling.

KEGG pathway enrichment analysis demonstrated that the identified targets were predominantly associated with "Chemokine signaling pathway", "ErbB signaling pathway" and "Viral protein interaction with cytokine and cytokine receptor" (Fig 2B). These enriched pathways collectively suggest that BaA may promote LUAD by disrupting chemokine-mediated immune regulation, oncogenic signaling activation, and cytokine receptor interactions.

### 3.3. Identification of core targets via machine learning algorithms

Next, we identified core target genes from 248 BaA-induced LUAD-related targets using LASSO, SVM-RFE and Random Forest algorithms. We first employed LASSO logistic regression with triple cross-validation penalty parameter tuning to identify 9 genes as potential core targets (Fig 3A–3B). The SVM-RFE algorithm selected 8 genes as high-priority candidates (Fig 3C). The Random Forest algorithm, which evaluates feature importance based on Gini impurity reduction, further out 6 candidates with importance scores > 10 (Fig 3D). Through a Venn diagram integrating results from three methods, we ultimately identified five definitive core targets associated with BaA-induced LUAD: *TNNC1, ABCC3, CRABP2, CXCL12*, and *OLR1* (Fig 3E). The triple-algorithm screening ensures high confidence in these core targets, which are likely key mediators of BaA-induced LUAD progression.

### 3.4. Differential expression analysis and validation of core targets using GEO datasets

We obtained three independent LUAD transcriptomic datasets from the GEO database: GSE10072, GSE32863, and GSE31210. After merging the datasets, removing batch effects, and performing quantile normalization, probe IDs were

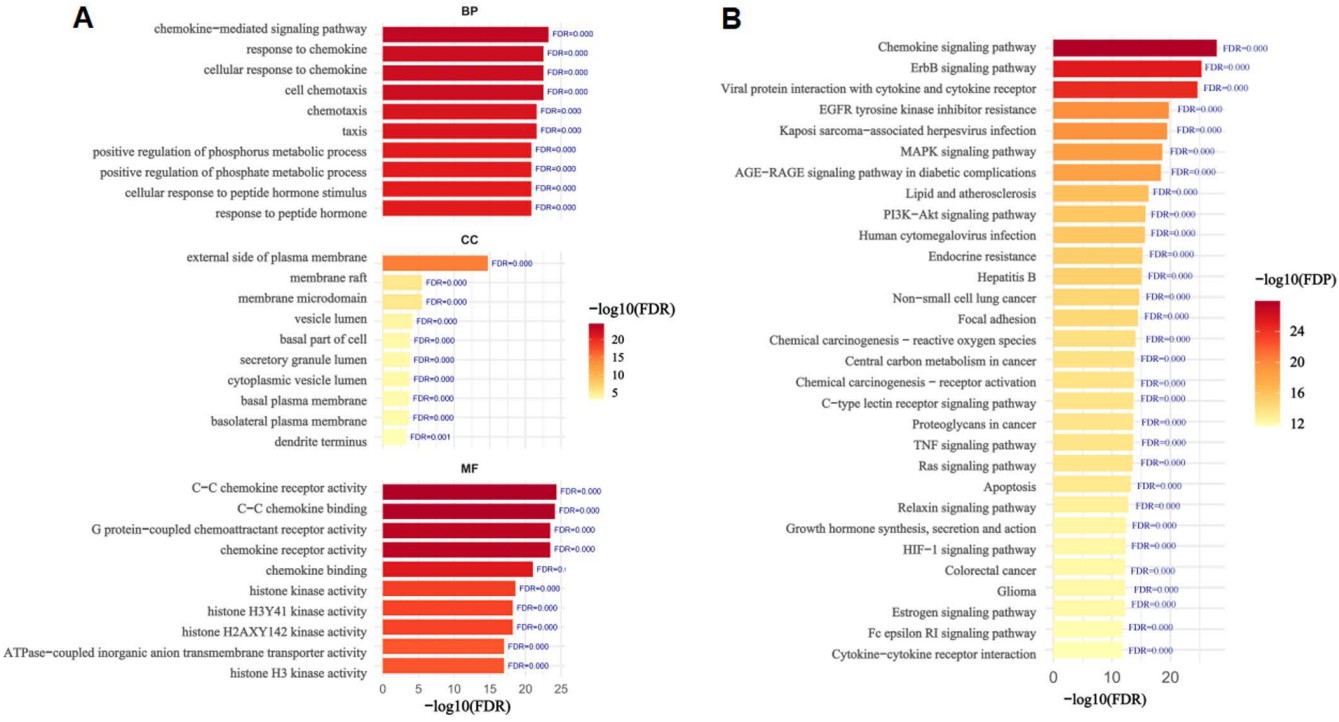

**Fig 2. GO and KEGG pathway enrichment analysis of BaA-induced LUAD-related intersection targets. (A)** GO functional enrichment analysis of BaA-induced LUAD-related intersection targets (n = 248) across BP, CC, and MF categories. The x-axis represents the number of enriched genes, y-axis represents the top 10 terms with the lowest false discovery rate (FDR). Color gradient indicates FDR values (darker = more significant enrichment, adjusted *p* < 0.05). **(B)** KEGG pathway enrichment analysis of BaA-induced LUAD-related intersection targets (n = 248). The x-axis represents enrichment ratio (target genes in pathway/ total genes in pathway), y-axis represents top enriched pathways. Bubble size indicates the number of enriched target genes; color gradient indicates FDR values (darker = more significant, adjusted *p* < 0.05).

converted to gene symbols and duplicate samples were removed, resulting in 10,266 unique genes. Differential expression analysis was conducted using the limma package, identifying 2145 differentially expressed genes (DEGs) with |log$_2$FC| > 1 and adjusted p-value, p < 0.05. The top 50 DEGs were visualized using volcano plots and heatmaps (Fig 4A–4B).

Subsequently, we validated the expression and diagnostic potential of five core targets. Compared with Normal samples, the expression of *ABCC3*, and *CRABP2*, was significantly upregulated in LUAD samples (*p* < 0.01), whereas *CXCL12*, *OLR1*, and *TNNC1* expression was markedly downregulated (*p* < 0.01). (Fig 4C). To assess their diagnostic efficacy, we constructed ROC curves and calculated the AUC. The AUC values for these five genes (*TNNC1, ABCC3, CRABP2, CXCL12*, and *OLR1*) were 0.886, 0.885, 0.844, 0.841, and 0.846, respectively (Fig 4D), indicating their strong potential as biomarkers for LUAD diagnosis and supporting their roles in disease pathogenesis. Collectively, these results validate that the five core targets are abnormally expressed in LUAD and possess high diagnostic value, confirming their biological relevance to BaA-induced LUAD.

### 3.5. Correlation analysis between core targets and immune cell infiltration in LUAD

Based on immune infiltration correlation analysis, the five core targets exhibited distinct immunomodulatory patterns in the lung adenocarcinoma (LUAD) microenvironment (Fig 5A–5B). *TNNC1* showed positive correlations with multiple anti-tumor immune cells, such as T cells CD8, T cells CD4 memory resting, NK cells resting and activated, and Monocytes.

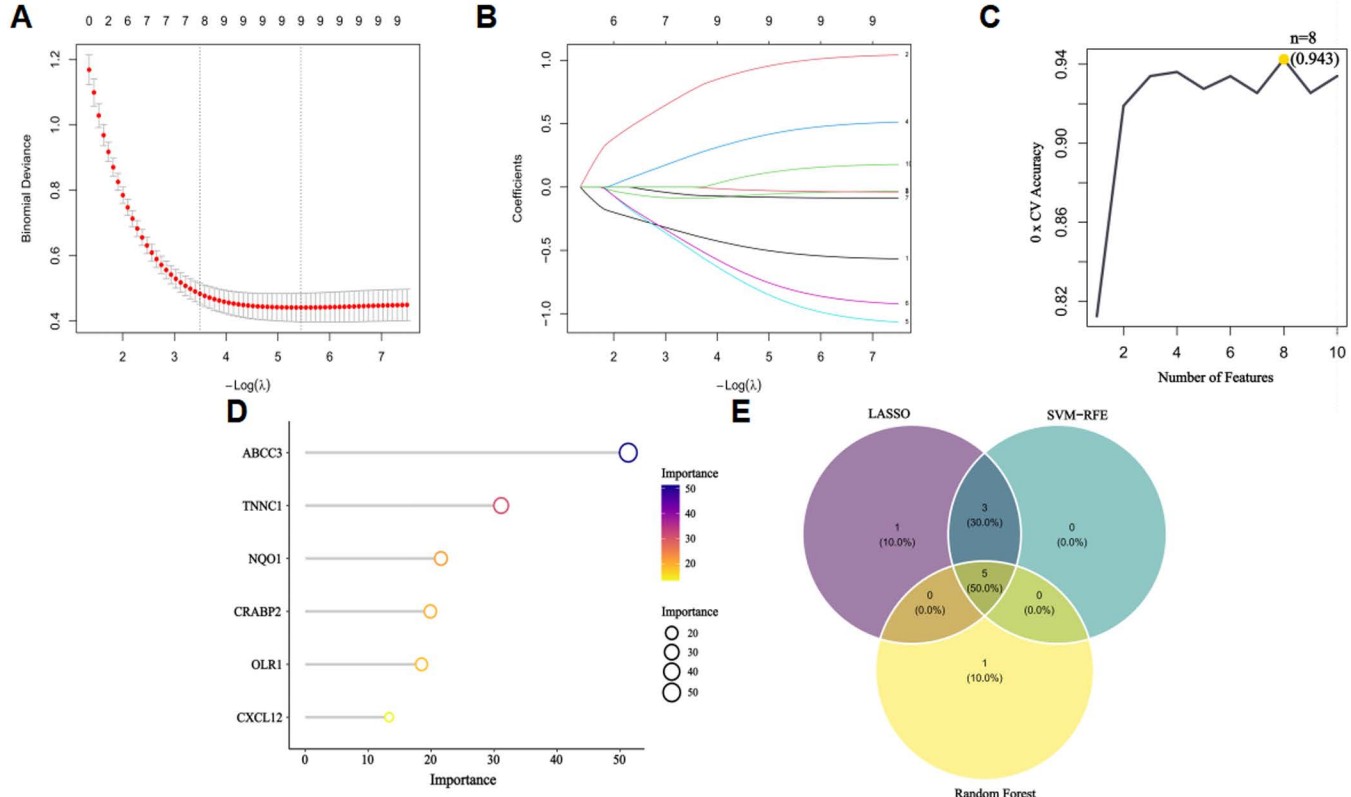

**Fig 3. Identification of core target genes for BaA-induced LUAD via machine learning algorithms. (A, B)** LASSO logistic regression: The abscissa represents the number of genes in the model corresponding to different λ values; 9 candidate core genes were identified at the minimum cross-validation λ. **(C)** Feature selection results of the SVM-RFE algorithm, with 8 candidate core genes identified. **(D)** Feature importance ranking via Random Forest algorithm, with 6 candidate core genes selected (importance score > 10). **(E)** Venn diagram showing the overlap of candidate core genes from LASSO, SVM-RFE, and Random Forest algorithms, identifying 5 definitive core targets for BaA-induced LUAD.

In contrast, it was negatively correlated with B cell subsets, T cells CD4 naive, T cells follicular helper, T cells regulatory (Tregs), and Macrophages M0/M1. *ABCC3, CRABP2, CXCL12*, and *OLR1*, however, were generally positively correlated with immunosuppressive cell types including Macrophages M0/M1, Tregs, and T cells follicular helper, while being negatively correlated with anti-tumor immune cells such as T cells CD8, T cells CD4 memory resting, and NK cells. Additionally, *CXCL12* and *OLR1* showed positive associations with Neutrophils, Eosinophils, and Monocytes. These findings reveal a dual immunomodulatory role of core targets, where *TNNC1* may enhance anti-tumor immunity while *ABCC3, CRABP2, CXCL12*, and *OLR1* promote immunosuppression, collectively remodeling the LUAD immune microenvironment.

### 3.6. Independent clinical validation of core targets using TCGA data via UALCAN

To further verify the expression patterns and clinical relevance of the five core targets, we performed independent clinical validation using TCGA-LUAD transcriptomic data accessed through the UALCAN platform. Consistent with the integrated GEO dataset analysis (GSE10072, GSE32863, GSE31210), the validation results showed significant dysregulation of all core targets in LUAD tissues compared to normal lung tissues ($p < 0.05$, Fig 6A–6E). Specifically, *ABCC3* and *CRABP2* were significantly upregulated in LUAD samples, while *TNNC1, CXCL12*, and *OLR1* exhibited marked downregulation, confirming the stability and reliability of the core targets' expression characteristics across different cohorts.

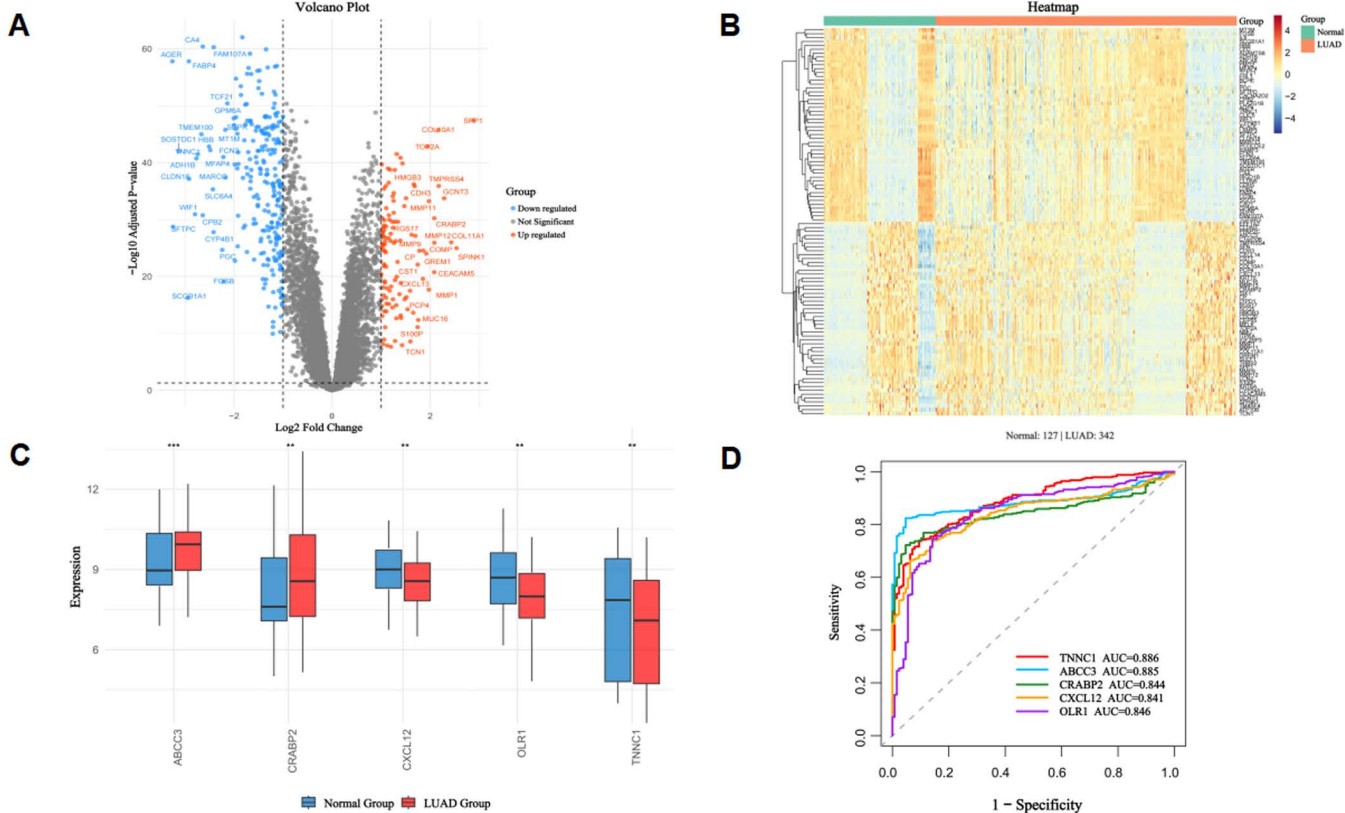

**Fig 4. Differential expression analysis and validation of core targets using integrated GEO datasets. (A)** Volcano plot of differentially expressed genes (DEGs) from integrated GEO datasets (GSE10072, GSE32863, GSE31210). The x-axis represents log$_2$ fold change (log$_2$FC) of gene expression (LUAD vs. normal samples), and the y-axis represents -log$_{10}$ (adjusted p-value). Red dots indicate upregulated DEGs (log$_2$|FC|>1, adjusted p<0.05), blue dots indicate downregulated DEGs (log$_2$|FC|>1, adjusted p<0.05), and gray dots indicate non-significantly differentially expressed genes. **(B)** Heatmap of the top 50 DEGs from integrated GEO datasets (GSE10072, GSE32863, GSE31210). Rows represent individual genes, columns represent samples (green=normal samples, red=LUAD samples). The color gradient (blue to red) indicates normalized gene expression levels (blue=low expression, red=high expression), with values standardized by z-score. DEGs were screened using the criteria: log$_2$|FC|>1 and adjusted p<0.05. **(C)** Expression levels of five core targets in normal and LUAD samples (*p<0.05, **p<0.01, ***p<0.001). Blue indicates normal group samples, while red represents LUAD group samples. **(D)** Diagnostic performance of five core targets in distinguishing normal and LUAD samples via ROC curves.

Subsequently, Kaplan-Meier survival analysis was conducted to explore the association between core target expression levels and overall survival (OS) of LUAD patients. The results revealed distinct prognostic trends (Fig 7A–7E).: *TNNC1* (*p*=0.28) and *CXCL12* (*p*=0.44) low-expression groups showed a tendency toward longer OS, and *OLR1* low-expression reached a near-statistically significant level (*p*=0.05) for improved survival. In contrast, *ABCC3* high-expression (*p*=0.05) and *CRABP2* high-expression (*p*=0.48) were associated with a trend toward poor prognosis. Notably, none of these associations achieved strict statistical significance (*p*>0.05), indicating that the expression levels of individual core targets may have limited independent prognostic value for LUAD patients. These findings collectively demonstrate that the core targets'expression patterns are consistent across GEO and TCGA cohorts, reinforcing their biological relevance to BaA-induced LUAD.

### 3.7. Molecular docking analysis of BaA with core targets for BaA-induced LUAD

As part of the network toxicology framework, we evaluated the interactions between BaA and five core targets through molecular docking analysis. The results showed that the binding affinities of BaA-CRABP2 (Fig 8A), BaA-OLR1 (Fig 8B), BaA-CXCL12 (Fig 8C), BaA-ABCC3 (Fig 8D), and BaA-TNNC1 (Fig 8E) were all below-5 kcal/mol, indicating their favorable

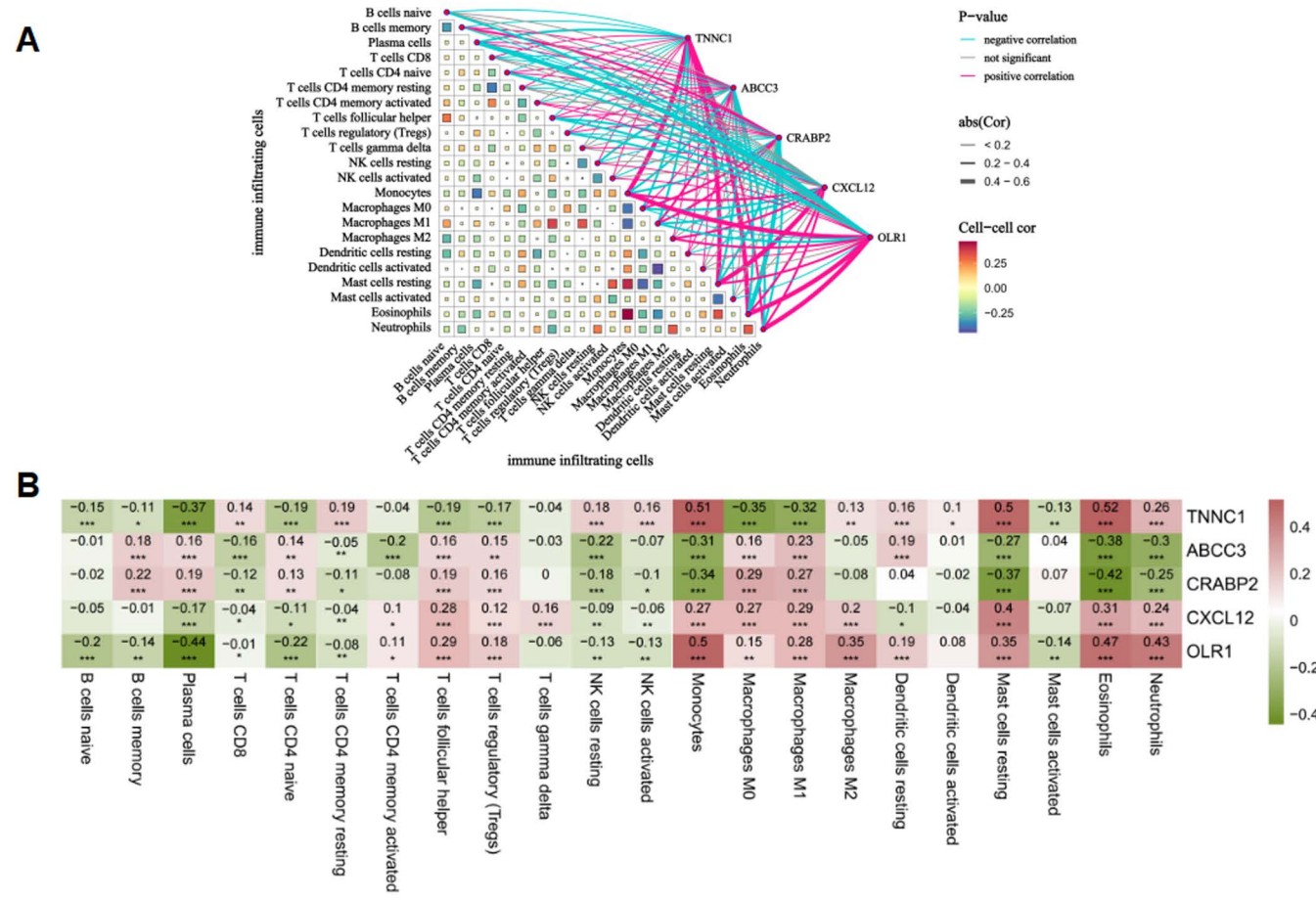

**Fig 5. Correlation between five core targets and immune cell infiltration in LUAD. (A)** Network diagram illustrating the correlation between core targets and immune cell types. **(B)** Correlation heatmap between five core targets and 22 immune cell types (from integrated GEO datasets). Color gradient indicates Pearson correlation coefficient (red = positive correlation, green = negative correlation). Asterisks denote statistical significance (*$p < 0.05$, **$p < 0.01$, ***$p < 0.001$).

binding potential [28]. Lower binding free energy values (i.e., more negative) from molecular docking correspond to stronger intermolecular forces and greater conformational stability between BaA (ligand) and the target proteins. A binding affinity lower than −7.0 kcal/mol is generally considered indicative of strong binding. Among the five targets, BaA showed the lowest binding affinity with *CRABP2* (−8.4 kcal/mol), suggesting the highest binding affinity for this target. It is worth noting that, due to the lack of hydrogen bond donors or acceptors in its structure, BaA primarily relies on characteristic hydrophobic interactions and π–π stacking to achieve stable binding within the hydrophobic pockets of the targets. These extensive surface contacts contribute to highly cumulative van der Waals forces, which stabilize the resulting complexes. The strong binding affinity between BaA and *CRABP2* (−8.4 kcal/mol) suggests *CRABP2* may be a key direct target of BaA in LUAD.

### 3.8. MD simulations and binding free energy calculation

100 ns MD simulations were performed using GROMACS 2021.4 (random seed: 123456) to validate BaA-core target binding stability. Docked outputs were processed via PyMOL (dehydration, hydrogenation) and saved as PDB (protein) and MOL2 (ligand) files. AMBER99SB-ILDN force field (protein) and Sobtop-generated parameters (BaA) were used, with TIP3P solvation, Na⁺ neutralization, and 10 Å box distance. After energy minimization (steepest descent/conjugate

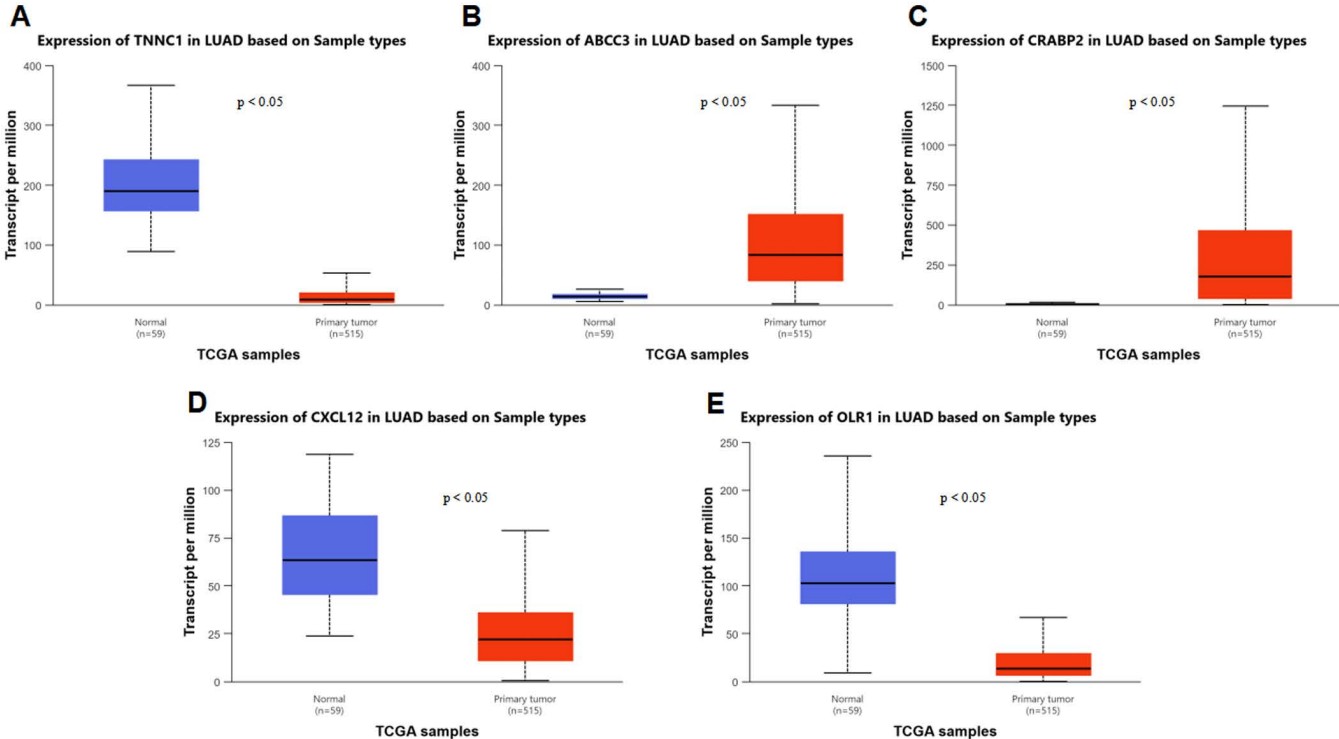

**Fig 6. Differential expression analysis of core targets in UALCAN database.** The UALCAN platform was used to compare mRNA expression levels of five core targets in 515 LUAD samples and 59 Normal samples from the TCGA database. **(A)** *TNNC1* expression in TCGA samples. **(B)** *ABCC3* expression in TCGA samples. **(C)** *CRABP2* expression in TCGA samples. **(D)** *CXCL12* expression in TCGA samples. **(E)** *OLR1* expression in TCGA samples. Statistical significance was assessed using Student's t-test (p-values shown in each figure).

gradient), conformational restraint, 100 ps NVT, and 200 ps NPT equilibration, production simulation ran with 2 fs step (50 million steps), LINCS-constrained H-bonds, 1.0 nm cutoffs, PME electrostatics, and V-rescale/Berendsen coupling (300 K, 1 bar). Trajectories (saved every 30 ps) were corrected (drift removal), and visualize with DulvyTools.

The structural stability and dynamics of the BaA-CRABP2 complex were analyzed over the 100 ns MD simulation trajectory. The Gibbs free energy landscape, projected along the first two principal components (PC1 and PC2), revealed a dominant low-energy basin, indicating a highly stable conformational state (Fig 9A). The root mean square deviation (RMSD) of the protein backbone converged, confirming the system reached equilibrium (Fig 9B). Consistent with this, the radius of gyration (Rg) and solvent-accessible surface area (SASA) profiles demonstrated sustained compactness and effective burial of the hydrophobic core (Fig 9C, 9D). Root mean square fluctuation (RMSF) analysis highlighted the flexibility of specific loop regions (Fig 9E). As BaA lacks hydrogen bond donors/acceptors, binding is primarily stabilized by hydrophobic interactions and π-π stacking within the *CRABP2* cavity; thus, hydrogen bonding was not a focus of the analysis. The stable conformational dynamics of the BaA-CRABP2 complex over 100 ns confirm the specific and robust interaction between BaA and *CRABP2*.

### 3.9. Construction of the AOP hypothesis framework

Building upon the methodology proposed by Daniel et al. [29] we developed a novel AOP hypothesis framework. This AOP proposes that BaA exposure may regulate the expression and activity of five core targets including *TNNC1, ABCC3, CRABP2, CXCL12*, and *OLR1*. Furthermore, it suggests that specific molecular functions such

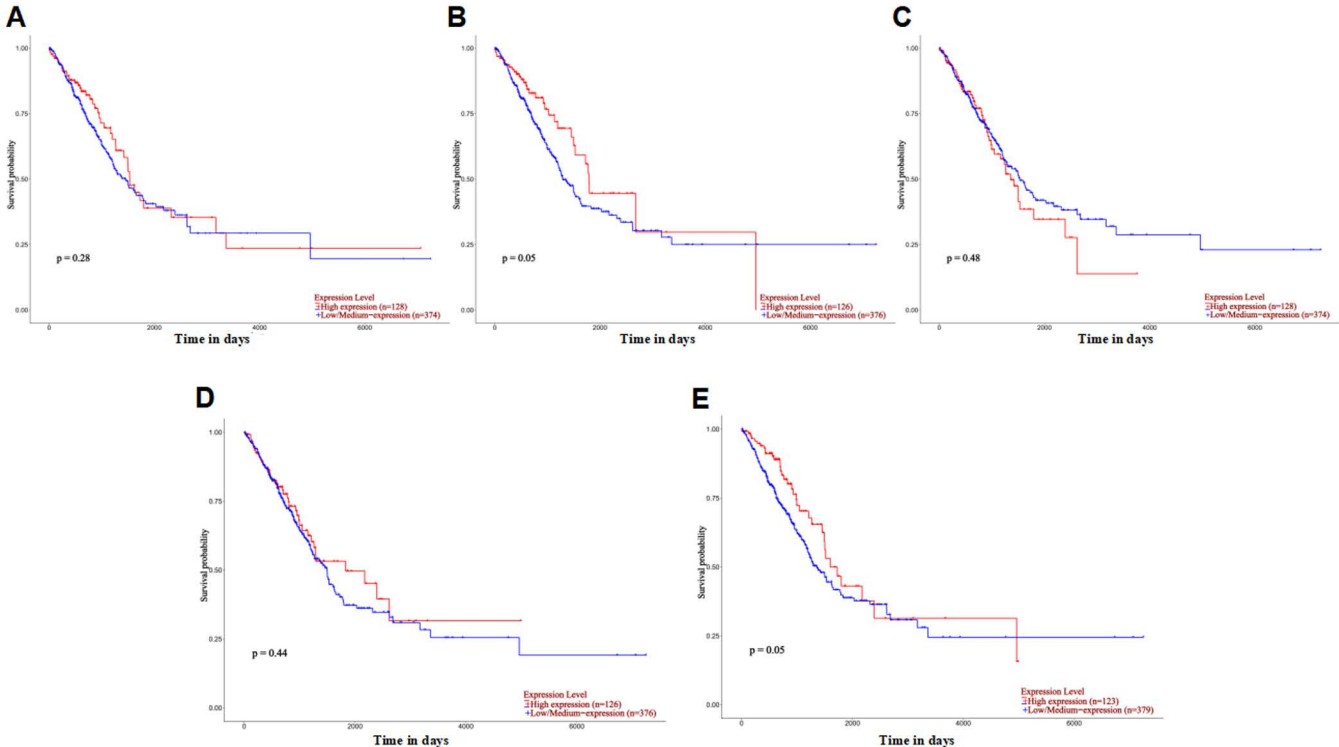

**Fig 7. Prognostic value of core targets in LUAD patients via Kaplan-Meier survival analysis (TCGA-LUAD cohort, accessed through UALCAN).** Kaplan-Meier curves (TCGA-LUAD cohort) stratified LUAD patients by median expression of core targets (High vs. Low/Medium groups). X-axis: survival time (days); Y-axis: survival probability. **(A)** Effect of *TNNC1* expression level on LUAD patient survival. **(B)** Effect of *ABCC3* expression level on LUAD patient survival. **(C)** Effect of *CRABP2* expression level on LUAD patient survival. **(D)** Effect of *CXCL12* expression level on LUAD patient survival. **(E)** Effect of *OLR1* expression level on LUAD patient survival. Statistical significance was assessed using Student's t-test (p-values shown in each figure). None of the targets showed statistically significant prognostic associations.

as "Chemokine signaling pathway", "ErbB signaling pathway", and "Viral protein interaction with cytokine and cytokine receptor" could contribute to pulmonary immune dysregulation and ultimately promote LUAD development (Fig 10). By systematically integrating these key genes and molecular functions, this study establishes a conceptual AOP that provides a theoretical basis for understanding BaA's potential role in LUAD pathogenesis. However, it should be emphasized that the proposed key event relationships in this AOP remain speculative and require further validation through direct experimental evidence. This AOP framework systematically links BaA exposure to LUAD development via core target regulation and pathway dysregulation, providing a standardized toxicological pathway for understanding BaA-induced carcinogenesis.

## 4. Discussion

BaA represents a prototypical environmental PAH contaminant with documented human exposure through multiple pathways including inhalation of combustion emissions and dietary intake. While previous studies have established BaA's genotoxic potential through DNA adduct formation [6], our integrated analysis reveals novel mechanistic insights into its role in LUAD pathogenesis, particularly through disruption of specific signaling networks and immune microenvironment modulation.

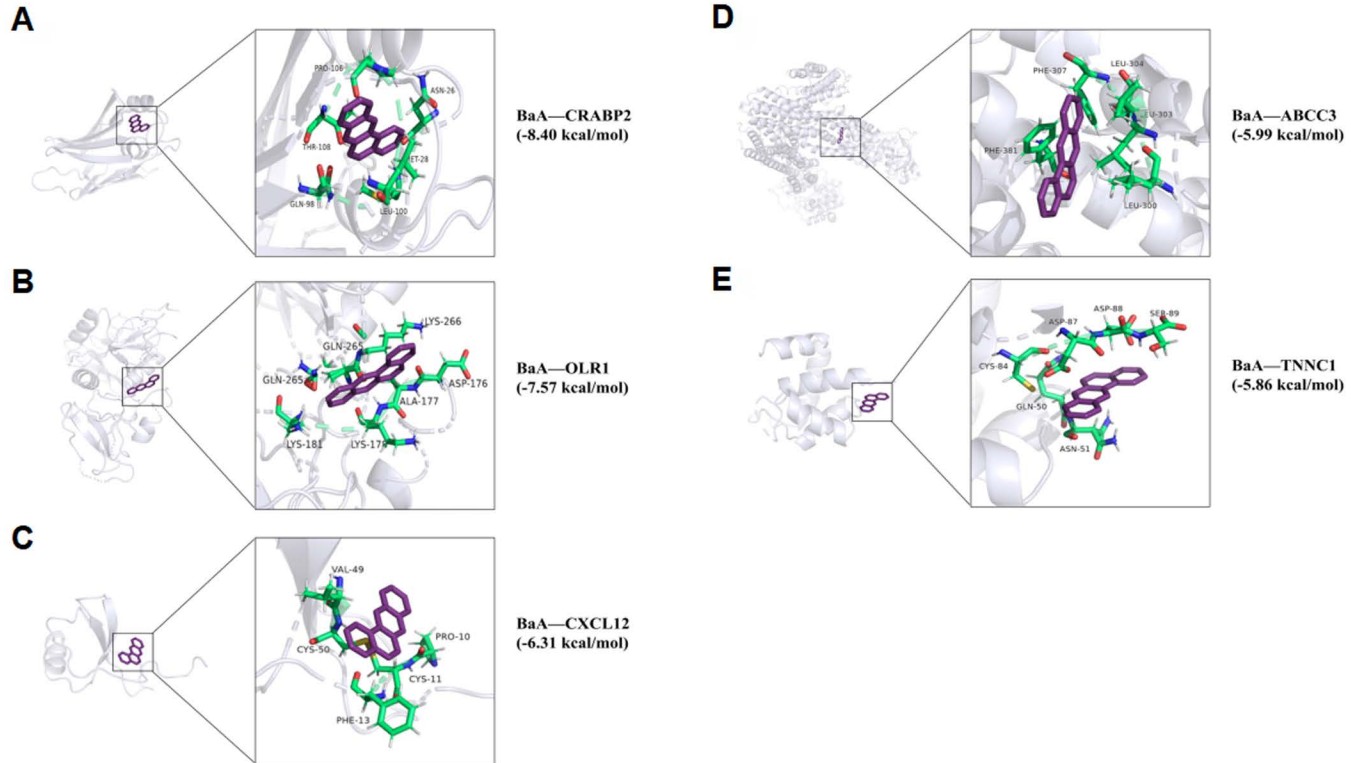

**Fig 8. Molecular docking results showing the binding affinity of BaA with five core targets for BaA-induced LUAD. (A)** BaA and Cellular retinoic acid binding protein 2 (CRABP2). **(B)** BaA and Oxidized low-density lipoprotein receptor 1 (OLR1). **(C)** BaA and C-X-C chemokine ligand 12 (CXCL12). **(D)** BaA and ATP binding cassette subfamily c member 3 (ABCC3). **(E)** BaA and Troponin c1, slow skeletal and cardiac type (TNNC1). Binding affinity values are provided to illustrate the strength of the interaction between BaA and each target.

## 4.1. Pathway dysregulation in BaA-induced LUAD

KEGG pathway analysis identified three significantly enriched pathways that collectively form a coordinated carcinogenic network. The Chemokine signaling pathway emerged as the most notably altered, with BaA potentially disrupting the homeostasis of the CXCL12-CXCR4 axis. This pathway not only facilitates tumor cell migration and invasion but also orchestrates the formation of an immunosuppressive microenvironment by recruiting myeloid-derived suppressor cells and regulatory T cells [30]. Specifically, aberrant chemokine signaling can promote angiogenesis through upregulation of pro-angiogenic factors while simultaneously inhibiting effective anti-tumor immunity [31].

The ErbB signaling pathway plays a critical role in the pathogenesis of BaA-induced LUAD. Key receptors in this pathway, such as EGFR and ErbB2, function as transmembrane tyrosine kinases and are frequently aberrantly activated in LUAD [32]. BaA metabolites, such as BaA-3,4-diol-1,2-epoxide, can modify the extracellular domain of EGFR, inducing its dimerization and autophosphorylation, thereby initiating downstream oncogenic signaling [33]. Sustained activation of the ErbB pathway further triggers the PI3K-AKT-mTOR and RAS-RAF-MEK-ERK cascades [34], driving uncontrolled cell proliferation, inhibiting apoptosis, and promoting tumor metastasis through epithelial-mesenchymal transition (EMT) [35].

Notably, the "Viral protein interaction with cytokine and cytokine receptor" pathway suggests that BaA may exploit viral mimicry mechanisms to disrupt immune surveillance. Environmental carcinogens can activate endogenous retroviral elements, triggering viral defense pathways that paradoxically create an inflammatory microenvironment conducive to tumor development [36]. This mechanism may explain the chronic inflammation observed in PAH-induced lung carcinogenesis.

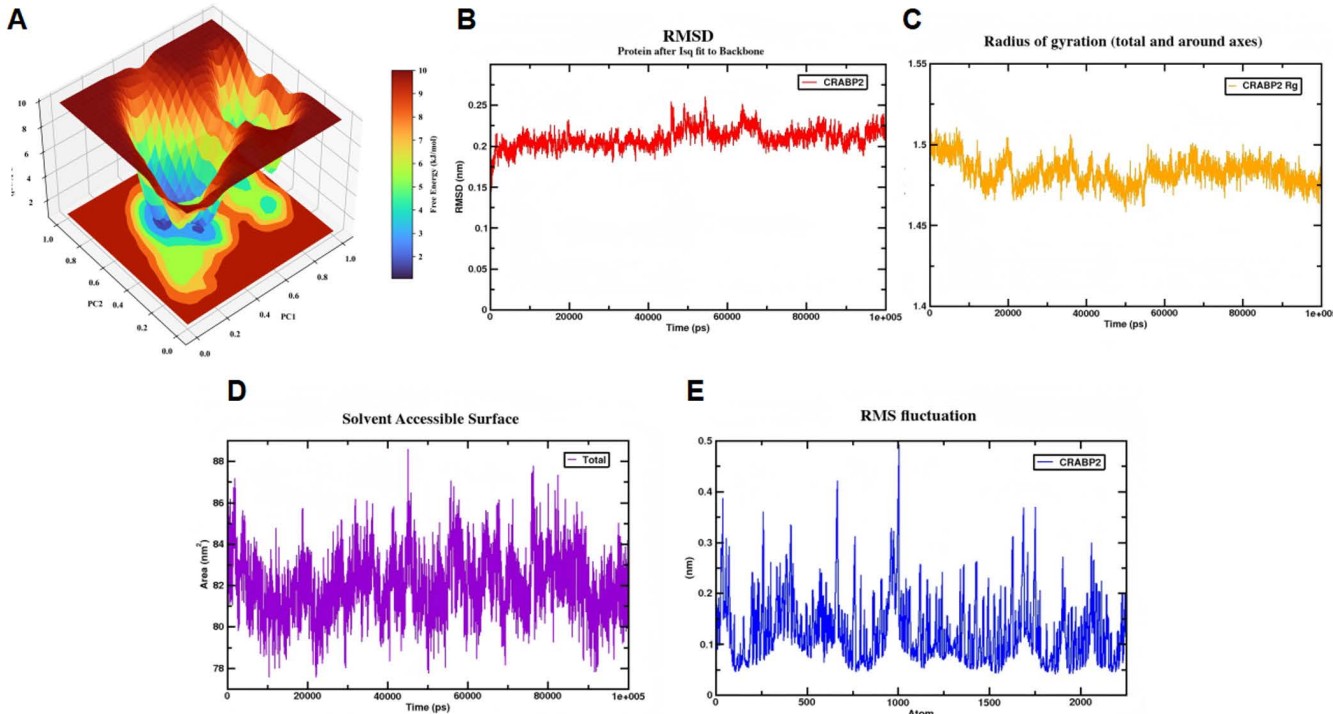

**Fig 9. Molecular dynamics (MD) simulations of the BaA-CRABP2 complex. (A)** Gibbs free energy landscape of the BaA-CRABP2 complex, showing the free energy distribution in the conformational space defined by the first two principal components (PC1, PC2). **(B)** RMSD values of the BaA-CRABP2 complex over time. **(C)** Rg values of the BaA-CRABP2 complex, indicating the compactness of the complex. **(D)** SASA values of the BaA-CRABP2 complex, illustrating the surface exposure. **(E)** RMSF values, showing the flexibility of the BaA-CRABP2 complex.

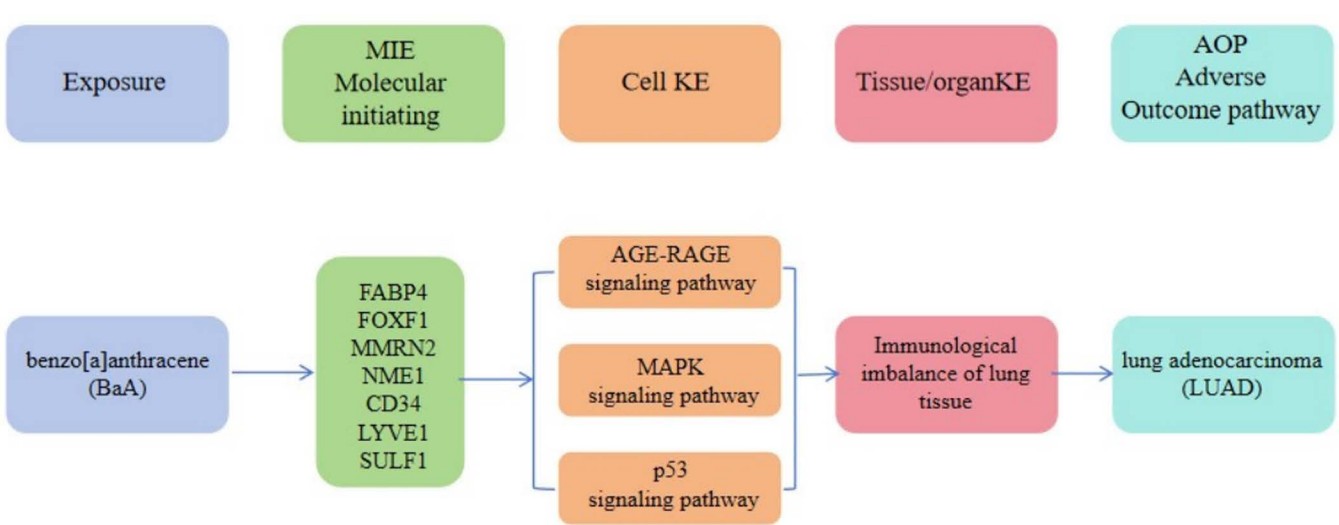

**Fig 10. Adverse outcome pathway (AOP) hypothesis framework for BaA-induced LUAD.**

## 4.2. Machine learning-established core targets: Functional significance and validation

The five core targets identified through our triple-algorithm approach represent functionally diverse yet interconnected players in LUAD pathogenesis. *TNNC1* demonstrated significant downregulation in LUAD samples, with emerging evidence supporting its tumor-suppressive role through calcium-mediated apoptosis regulation [37,38]. *ABCC3* is significantly upregulated in LUAD and functions as a broad-spectrum efflux transporter for both xenobiotics and endogenous metabolites. Its overexpression represents an adaptive response to chemical stress, aligning with BaA's role as an environmental stressor [39]. Through metabolite export, *ABCC3* not only confers multidrug resistance but also shapes a protumorigenic microenvironment, contributing to aggressive LUAD phenotypes [40]. *CRABP2* showed the strongest binding affinity with BaA. This protein regulates retinoic acid intracellular trafficking and signaling, with overexpression leading to pro-proliferative and anti-apoptotic effects through altered retinoic acid receptor activation [41]. BaA's high-affinity interaction with *CRABP2* suggests potential disruption of retinoid signaling, a key pathway in lung epithelial differentiation and carcinogenesis [42]. *CXCL12* downregulation in LUAD may disrupt its autocrine/paracrine signaling axis with CXCR4, potentially impairing tumor cell survival signals and organ-specific metastatic tropism [43]. Furthermore, reduced *CXCL12* levels could alter immune cell recruitment, possibly diminishing the immunosuppressive microenvironment and enhancing immune surveillance [44]. Similarly, *OLR1* downregulation in LUAD suggests a possible alteration in lipid-mediated inflammatory pathways. Lower *OLR1* expression may reduce oxidized LDL uptake and subsequent NF-κB activation, thereby attenuating chronic inflammation and creating a less supportive tumor microenvironment [45]. These targets demonstrate good diagnostic performance (AUC: 0.841–0.886 across three independent cohorts), underscoring their clinical relevance and confirming their fundamental role in the pathogenesis of BaA-induced LUAD.

## 4.3. Immune microenvironment remodeling by BaA

Our immune infiltration analysis reveals the complex immunomodulatory effects mediated by core targets of BaA. The positive correlation between *TNNC1* and CD8$^+$T cells as well as resting CD4$^+$memory T cells suggests that it may enhance anti-tumor immunity through calcium signaling-mediated T cell activation [38]. The downregulation of this target may consequently impair immune surveillance and promote tumor immune escape. In contrast, *ABCC3, CRABP2, CXCL12*, and *OLR1* collectively indicate an immunosuppressive microenvironment, manifested by broad positive correlations with regulatory T cells (Tregs), follicular helper T cells, and M0/M1 macrophages, while showing significant negative correlations with various anti-tumor lymphocytes. Notably, the promyeloid cell activity demonstrated by *CXCL12* and *OLR1* further suggests their critical role in recruiting immunosuppressive myeloid cell populations. In summary, these findings not only indicate that *TNNC1* may possess anti-tumor immunostimulatory functions, while *ABCC3, CRABP2, CXCL12*, and *OLR1* collaboratively shape an immunosuppressive microenvironment, but also reveal the dual mechanism of BaA in lung cancer progression from an immunoregulatory perspective: simultaneously activating oncogenic signaling pathways while systematically suppressing anti-tumor immune function.

## 4.4. Clinical relevance and prognostic implications of core targets

Independent clinical validation using TCGA-LUAD data via UALCAN confirmed the significant dysregulation of the five core targets, reinforcing their consistency across cohorts. These stable expression patterns underscore their biological relevance to BaA exposure and LUAD pathogenesis. Survival analysis revealed distinct prognostic trends, albeit without reaching strict statistical significance. *TNNC1* and *CXCL12* downregulation trended towards better survival, while high expression of *ABCC3* and *CRABP2* was associated with poorer outcomes. These findings suggest that while these targets are associated with disease, their independent prognostic value for patient stratification may be limited. Their primary clinical utility may reside as complementary indicators within a broader predictive panel.

### 4.5. Molecular docking and MD simulation: Binding mechanisms of BaA with core targets

Molecular docking revealed BaA's distinctive binding mode, characterized by predominant hydrophobic interactions and π-π stacking rather than conventional hydrogen bonding. This pattern reflects BaA's highly hydrophobic aromatic structure, which favors burial within protein hydrophobic pockets [46]. The exceptional binding affinity with *CRABP2* (−8.4 kcal/mol) stems from optimal complementarity with *CRABP2*'s hydrophobic ligand-binding cavity, which normally accommodates retinoids [47].

MD simulations confirmed the stability of BaA-CRABP2 complex formation, with favorable RMSD, Rg, and SASA profiles throughout the 100 ns trajectory. The entropy-driven binding, mediated primarily by hydrophobic effects, represents a characteristic interaction pattern for PAHs with their protein targets. This stable interaction suggests BaA may competitively inhibit retinoic acid binding, disrupting normal retinoid signaling and promoting carcinogenesis.

### 4.6. Integrated AOP hypothesis framework and public health implications

The AOP hypothesis framework proposed in this study systematically connects MIEs (BaA exposure) with the adverse outcome (LUAD development) through KEs including signaling pathway dysregulation and immune microenvironment remodeling. This framework not only provides a mechanistic basis for understanding BaA's carcinogenic potential but also identifies potential intervention points. Given the widespread presence of BaA in environmental and dietary sources, along with the multidimensional carcinogenic mechanisms revealed in this study, there is an urgent need to establish a more comprehensive risk assessment system. The constructed AOP framework establishes a standardized "exposure-effect" chain for BaA-induced LUAD, which facilitates the integration of existing toxicological evidence and provides theoretical support for advancing environmental health risk assessment of BaA.

### 4.7. Limitations and future perspectives

Notably, pollutant synergism represents a critical aspect of environmental carcinogenesis [48], as BaA typically coexists with other PAHs including benzo[a]pyrene, heavy metals such as arsenite, and volatile organic compounds in real world exposure scenarios [49]. Emerging evidence indicates these mixtures induce more severe genotoxicity and oncogenic pathway activation than individual compounds alone [50,51]. Specifically, arsenite can inhibit nucleotide excision repair of BaA induced DNA adducts while other co pollutants may exacerbate pro inflammatory immune microenvironment remodeling [52,53]. These synergistic interactions could amplify both the binding affinity of BaA to core targets identified in our study and the associated immune correlations, underscoring the necessity to consider complex pollutant mixtures in future environmental risk assessment frameworks for LUAD.

While our integrated research methodology provides comprehensive insights, several limitations must be acknowledged. First, computational predictions require experimental validation through in vitro and in vivo models, which is a critical step to refine research conclusions and enhance scientific rigor. Second, real-world BaA exposure typically forms complex mixtures with other PAHs and pollutants, potentially generating synergistic effects unobservable through single-compound analysis [54]. Future studies should incorporate toxicity assessments of these mixtures and conduct in-depth dose-response relationship investigations to strengthen the practical significance of risk assessments. Additionally, the precise temporal sequence of KEs in BaA-induced LUAD pathogenesis requires longitudinal studies. Large-scale, long-term prospective epidemiological studies are needed to systematically track the dynamic relationship between BaA exposure levels and LUAD incidence rates..

## 5. Conclusion

This study systematically elucidated the molecular mechanisms of BaA-induced LUAD by integrating network toxicology, machine learning, and computational simulations. A total of 248 candidate targets were successfully screened, from

which five core targets including *TNNC1, ABCC3, CRABP2, CXCL12*, and *OLR1* were prioritized via machine learning algorithms. Molecular docking and dynamics simulations demonstrated stable binding interactions between BaA and these targets, with the most prominent binding observed with *CRABP2*. Based on these findings, we constructed an AOP theoretical framework for BaA-induced LUAD, systematically illustrating the toxicological pathway from MIEs to adverse outcomes. The integrated analytical strategy established in this study not only overcomes the limitations of traditional single target research but also provides new methodological and theoretical foundations for health risk assessment of environmental pollutants.

## Abbreviations

| Abbreviations | Full Term | First use section |
|---|---|---|
| ABCC3 | ATP Binding Cassette Subfamily C Member 3 | Results |
| AGE-RAGE | Advanced Glycation End Product-Receptor for Advanced Glycation End Products | Introduction |
| AOP | Adverse Outcome Pathway | Abstract |
| AUC | Area Under the Curve | Materials and Methods |
| BaA | Benzo[a]anthracene | Abstract |
| BP | Biological Process | Results |
| CC | Cellular Component | Results |
| ChEMBL | Chemical Biology Database | Materials and Methods |
| CID | Compound Identifier | Materials and Methods |
| CRABP2 | Cellular Retinoic Acid Binding Protein 2 | Results |
| CTD | Comparative Toxicogenomics Database | Materials and Methods |
| CXCL12 | C-X-C Motif Chemokine Ligand 12 | Results |
| CYP1A1 | Cytochrome P450 Family 1 Subfamily A Member 1 | Introduction |
| CYP1B1 | Cytochrome P450 Family 1 Subfamily B Member 1 | Introduction |
| DAVID | Database for Annotation, Visualization and Integrated Discovery | Materials and Methods |
| DEG | Differentially Expressed Gene | Materials and Methods |
| EGFR | Epidermal Growth Factor Receptor | Discussion |
| EMT | Epithelial-Mesenchymal Transition | Discussion |
| ErbB | Erythroblastic Leukemia Viral Oncogene Homolog | Results |
| FABP4 | Fatty Acid Binding Protein 4 | Abstract |
| FC | Fold Change | Materials and Methods |
| FDR | False Discovery Rate | Results |
| GAFF | General Amber Force Field | Materials and Methods |
| GeneCards | GeneCards Database | Materials and Methods |
| GEO | Gene Expression Omnibus | Materials and Methods |
| GO | Gene Ontology | Materials and Methods |
| GSVA | Gene Set Variation Analysis | Materials and Methods |
| KEGG | Kyoto Encyclopedia of Genes and Genomes | Materials and Methods |
| KEs | Key Events | Materials and Methods |
| LASSO | Least Absolute Shrinkage and Selection Operator | Abstract |
| LUAD | Lung Adenocarcinoma | Abstract |
| MD | Molecular Dynamics | Materials and Methods |
| MF | Molecular Function | Results |
| MIEs | Molecular Initiating Events | Materials and Methods |
| NF-κB | Nuclear Factor Kappa B | Discussion |
| OLR1 | Oxidized Low Density Lipoprotein Receptor 1 | Results |

| Abbreviations | Full Term | First use section |
|---|---|---|
| OMIM | Online Mendelian Inheritance in Man | Materials and Methods |
| PAHs | Polycyclic Aromatic Hydrocarbons | Abstract |
| PC | Principal Component | Results |
| PDB | Research Collaboratory for Structural Bioinformatics Protein Data Bank | Materials and Methods |
| PI3K-AKT-mTOR | Phosphoinositide 3-Kinase-Protein Kinase B-Mammalian Target of Rapamycin | Discussion |
| PubChem | PubChem Database | Materials and Methods |
| RAS-RAF-MEK-ERK | Rat Sarcoma Virus-Rapidly Accelerated Fibrosarcoma-Mitogen-Activated Protein Kinase Kinase-Extracellular Signal-Regulated Kinase | Discussion |
| RESP | Restrained Electrostatic Potential | Materials and Methods |
| Rg | Radius of Gyration | Results |
| RMSD | Root Mean Square Deviation | Results |
| RMSF | Root Mean Square Fluctuation | Results |
| ROC | Receiver Operating Characteristic | Materials and Methods |
| ROS | Reactive Oxygen Species | Introduction |
| SASA | Solvent-Accessible Surface Area | Results |
| SLC1A5 | Solute Carrier Family 1 Member 5 | Introduction |
| SMILES | Simplified Molecular Input Line Entry System | Materials and Methods |
| ssGSEA | Single-Sample Gene Set Enrichment Analysis | Materials and Methods |
| SULF1 | Sulfatase 1 | Abstract |
| SVM-RFE | Support Vector Machine-Recursive Feature Elimination | Materials and Methods |
| TNNC1 | Troponin C1, Slow Skeletal and Cardiac Type | Results |
| Tregs | Regulatory T Cells | Results |
| TTD | Therapeutic Target Database | Materials and Methods |
| UniProt | Universal Protein Resource | Materials and Methods |

## Supporting information

**S1 Table. BaA-related targets.**
(XLSX)

**S2 Table. LUAD-related targets.**
(XLSX)

**S3 Table. Intersection targets.**
(XLSX)

## Author contributions

**Conceptualization:** Zhiyao Shi, Xixing Wang.

**Data curation:** Zhiyao Shi, Zhiyong Fang, Qiang Qin.

**Formal analysis:** Zhiyao Shi, Zhiyong Fang, Qiang Qin.

**Funding acquisition:** Xixing Wang.

**Methodology:** Zhiyao Shi, Yu Gao, Xixing Wang.

**Project administration:** Yu Gao, Likun Liu.

**Resources:** Yu Gao, Xi Yang.

**Software:** Zhiyao Shi, Xi Yang.

**Supervision:** Xi Yang, Likun Liu, Xixing Wang.

**Validation:** Yu Gao, Xi Yang, Likun Liu.

**Visualization:** Zhiyao Shi, Xi Yang, Likun Liu.

**Writing – original draft:** Zhiyao Shi, Zhiyong Fang, Qiang Qin.

**Writing – review & editing:** Xixing Wang.

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
