## [Decision Letter · Decision Letter 0]

14 Nov 2025

Dear Dr. WANG,

Thank you for submitting your manuscript to PLOS ONE. After careful consideration, we feel that it has merit but does not fully meet PLOS ONE’s publication criteria as it currently stands. Therefore, we invite you to submit a revised version of the manuscript that addresses the points raised during the review process.

We look forward to receiving your revised manuscript.

Kind regards,

Jiafu Li

Academic Editor

PLOS ONE

Journal Requirements:

3. Please note that PLOS One has specific guidelines on code sharing for submissions in which author-generated code underpins the findings in the manuscript. In these cases, we expect all author-generated code to be made available without restrictions upon publication of the work. Please review our guidelines at https://journals.plos.org/plosone/s/materials-and-software-sharing#loc-sharing-code and ensure that your code is shared in a way that follows best practice and facilitates reproducibility and reuse.

This work was supported by a grant from the Scientific research project funded by State Administration of Traditional Chinese Medicine of the People's Republic of China (2022-245,202203,2018-131);Department of Science and Technology of Shanxi Province (02103021224437,[2019]61);Health Commission of Shanxi Provincial(2020TD04,2022XM10)

This work was supported by a grant from the Scientific research project funded by State Administration of Traditional Chinese Medicine of the People's Republic of China (2022-245,202203,2018-131);Department of Science and Technology of Shanxi Province (02103021224437,[2019]61);Health Commission of Shanxi Provincial(2020TD04,2022XM10) The authors would like to thank the public databases including GEO, PDB, and PubChem for providing data support, and acknowledge the team members for their assistance in data organization and figure optimization.

This work was supported by a grant from the Scientific research project funded by State Administration of Traditional Chinese Medicine of the People's Republic of China (2022-245,202203,2018-131);Department of Science and Technology of Shanxi Province (02103021224437,[2019]61);Health Commission of Shanxi Provincial(2020TD04,2022XM10)

7. Please amend either the abstract on the online submission form (via Edit Submission) or the abstract in the manuscript so that they are identical.

8. Please ensure that you refer to Figure 9 in your text as, if accepted, production will need this reference to link the reader to the figure.

9. Please upload a new copy of Figure 1, 2, 3, 4, as the detail is not clear. Please follow the link for more information: https://journals.plos.org/plosone/s/figures

Reviewers' comments:

Reviewer's Responses to Questions

**Comments to the Author**

1. Is the manuscript technically sound, and do the data support the conclusions?

Reviewer #1: Partly

Reviewer #2: Yes

Reviewer #3: Partly

Reviewer #4: Partly

Reviewer #5: Yes

2. Has the statistical analysis been performed appropriately and rigorously?

Reviewer #1: Yes

Reviewer #2: Yes

Reviewer #3: Yes

Reviewer #4: No

Reviewer #5: No

3. Have the authors made all data underlying the findings in their manuscript fully available?

Reviewer #1: Yes

Reviewer #2: Yes

Reviewer #3: No

Reviewer #4: No

Reviewer #5: Yes

4. Is the manuscript presented in an intelligible fashion and written in standard English?

Reviewer #1: No

Reviewer #2: Yes

Reviewer #3: Yes

Reviewer #4: Yes

Reviewer #5: Yes

Reviewer #1: This manuscript meets PLOS ONE’s basic publication standards and is recommended for acceptance after major revision. This study, which explores the toxicological association between the environmental pollutant Benzo[a]anthracene (BaA) and lung adenocarcinoma (LUAD) via an interdisciplinary approach integrating network toxicology, machine learning, bioinformatics, and computational simulations, addresses a critical topic in public health and environmental toxicology with clear scientific significance.

However, there are some revisions Needed:

1. Lack of In Vitro/In Vivo Functional Validation: Core conclusions rely on bioinformatics and computational simulations without validation via in vitro cell experiments or in vivo models; suggest supplementing in vitro experiments for 1–2 key targets.

2. Single Dataset Limitation: Core target validation relies solely on GSE10072 (sample size unspecified); suggest adding 1–2 independent datasets for validation and specifying GSE10072’s sample size.

3. Ignoring Pollutant Synergistic Effects: BaA’s coexistence with other pollutants and their synergistic carcinogenicity are unaddressed; suggest expanding the "Discussion" and "Limitation" section to discuss or analyze interactions if relevant data exist.

4. Incomplete Figure Details: Some figures lack key details (e.g., Venn diagram target counts, GO/KEGG plot FDR values, molecular docking critical amino acid residues); suggest supplementing these to let readers obtain core data directly from figures.

5. Unified Terminology: Inconsistent abbreviations (e.g., "Benzo[a]anthracene" vs. "BaA", "epithelial-mesenchymal transition" vs. "EMT") reduce readability; suggest defining abbreviations on first use (e.g., "Benzo[a]anthracene (BaA)") and standardizing usage.

In conclusion, This manuscript meets PLOS ONE’s publication standards with rigorous design, clear significance, and sufficient data support; addressing the above revisions (supplementing validation, refining figures, enhancing data accessibility) will significantly improve its persuasiveness, so a "Major Revision" decision is recommended.

Reviewer #2: This manuscript integrates network toxicology, machine learning, and multi-dimensional bioinformatics, complemented by molecular docking and 100-ns MD, to propose multi-target mechanisms for BaA in LUAD and an AOP framework. The narrative is coherent and the results are largely consistent (seven core genes with high AUCs), giving methodological and risk-assessment value. I recommend minor revision.

1. Unify chemical nomenclature to “benzo[a]anthracene (BaA)” across title and Methods. Correct “BPS” to “BaA” in Section 2.1 and keep PubChem CID consistent throughout. Also fix the Fig. 5 caption (“five core targets”) to “seven”.

2. PLOS ONE requires open access to underlying data and analysis scripts. “Available upon reasonable request” is not compliant. Please deposit all R/Python scripts and parameter files (with session info/versions) to an open repository (e.g., Zenodo/OSF; GitHub mirrored to Zenodo) and provide a DOI in the Data Availability Statement.

3. Please add full reproducibility details for docking/MD: PDB IDs/chains and preprocessing; Vina grid center/size, exhaustiveness, replicates, scoring/version; MD time step, constraints, cutoffs, thermostat/barostat, trajectory interval, and exact equilibration/production settings; plus random seeds and software versions. The current text lists tools but lacks parameters.

4. Given the small size of GSE10072 and near-perfect single-gene AUCs, report sample sizes and provide ROC 95%CIs with resampling. Add independent validation cohorts (e.g., TCGA-LUAD, GSE31210/GSE72094) and consider multi-marker models with calibration/decision-curve analysis to mitigate overfitting. Document these steps in Methods/Results.

5. To enhance the connection between the BaA (PAH) chemical background, ROS/¹O₂ related discussions and methodologies, it is recommended to supplement the following literature in the introduction/discussion: 10.1002/cjoc.202300637; 10.1002/cjoc.202400602; 10.1002/cjoc.202300614.

Reviewer #3: Overall, in this study, the authors explored the toxicological mechanism of BaA in lung adenocarcinoma through machine learning and multi-dimensional bioinformatics, which is commendable. However, the description of the core targets’ expression in lung adenocarcinoma does not match the data shown in the figures, which causes a fatal impact on the subsequent analyses. Overall, part of the results are unreliable.

1. In the manuscript, there is a problem with the inconsistent use of abbreviations. Abbreviations such as LUAD, BaA, PAHs, BPS, and AOP are either redefined multiple times after the first definition or used without any definition.

2. In section 3.2 “Functional enrichment analysis of BaA-induced LUAD,” there are obvious writing errors. For example, in phrases like "gland development," "epithelial cell proliferation," the quotation marks are misplaced.The comma should be placed outside the quotation marks.

3. In section 2.4 “Screening of Core Targets by Machine Learning,” the authors mention using two machine learning models (both classification-based models) to identify core targets. However, it is unclear how the models determine which genes are the core targets and on what basis. This section should include a detailed description of how the training data were prepared.

4. The gene name formatting also seems problematic. I believe the gene symbols should be italicized, as they represent genes rather than proteins.

5. In section 3.5 “Verify the expression and diagnostic effectiveness of seven key targets in lung adenocarcinoma,” Figure 5B shows extremely high (almost perfect) AUC values, such as 0.998 or 0.996, which I believe are classic warning signs of overfitting. It is unclear on which dataset these ROC curves (Figure 5B) were plotted.

6. In section 3.5 “Verify the expression and diagnostic effectiveness of seven key targets in lung adenocarcinoma,” the authors state that “Compared with normal tissues, lung adenocarcinoma lesions showed significantly elevated expression of FABP4, FOXF1, MMRN2, NME1, CD34, LYVE1, and SULF1 (Fig. 5A).” However, upon examining Figure 5A closely, I found that only NME1 and SULF1 are upregulated, while the others are downregulated. I believe this major inconsistency between the text and data (Figure 5A) critically undermines the validity of the subsequent analyses. The impact is reflected at several levels:

(1) Impact on diagnostic effectiveness (AUC): probably minimal. This part of the analysis may still hold, because ROC curves and AUC values measure the ability of a marker to distinguish between “patients” and “controls,” regardless of the direction of change. A gene significantly downregulated in cancer (e.g., AUC = 0.99) and one significantly upregulated in cancer (e.g., AUC = 0.99) can both serve as excellent diagnostic biomarkers. Therefore, even if FABP4, FOXF1, etc., are downregulated, they may still show the extremely high AUC values (e.g., 0.996, 0.998) reported in the manuscript. Moreover, the molecular docking (Section 3.7) and molecular dynamics simulation (Section 3.8) analyses are only minimally affected.

(2) However, the impact on immune cell infiltration analysis (Section 3.6) and AOP framework construction (Section 3.9) is fatal. In Section 3.6, the authors’ logic assumes that these seven core genes are upregulated in cancer, and that certain “pro-cancer” immune cells (such as M1 macrophages) are also increased in cancer. The authors then claim that the expression of these seven genes is positively correlated with M1 macrophages, concluding that the upregulated genes drive M1 macrophage infiltration and thereby promote cancer — a coherent story. However, my finding (Figure 5A) indicates that most of these seven genes are actually downregulated in cancer, while M1 macrophages are reported as upregulated. If both remain “positively correlated” under these conditions (i.e., gene downregulation coincides with M1 increase), this directly contradicts the stated M1 upregulation.

In Section 3.9, the AOP (Adverse Outcome Pathway) is presented as the theoretical foundation summarizing the entire study. However, this framework is based on a false premise that BaA induces upregulation of all seven genes, subsequently activating the p53/MAPK pathway and ultimately leading to LUAD. Yet, my observation shows that five of these genes are downregulated and only two are upregulated, meaning the authors’ proposed AOP mechanism is fundamentally flawed. The “theoretical foundation” they constructed is inconsistent with their own data (Figure 5A).

Reviewer #4: The manuscript by Shi et al. investigates the toxicological mechanisms of Benzo[a]anthracene (BaA) exposure and its relationship with lung adenocarcinoma (LUAD) through an integrated systems-level approach. The authors employ various computational and bioinformatics analyses, including network toxicology, machine learning, and molecular docking/dynamics simulations. While the analyses are extensive, the study relies entirely on public databases and predictions, without any experimental validation to support the proposed mechanisms. Additionally, several important issues should be addressed to improve the accuracy of the manuscript.

Specific Comments

1.Line 58: A reference is missing to support the statement regarding serum BaA levels in LUAD patients.

2.Figure 2B: The KEGG pathway enrichment includes “pathways in cancer” and multiple general cancer terms. Since this study focuses specifically on LUAD, the enrichment pattern appears overly broad, and it is unclear why no immune-related pathways are highlighted. The authors should clarify or refine the enrichment criteria.

3.Figure 4: The authors used the GSE10072 dataset for expression validation. This microarray dataset is relatively small and outdated compared to the TCGA-LUAD RNA-seq cohort. To reduce bias and enhance reliability, the authors should validate their results using at least one larger RNA-seq dataset.

4.Figure 5A: The text states that MMRN2 is enriched in the treatment group; however, in the figure, MMRN2 appears higher in the control group. Please verify and correct this discrepancy.

5.Section 3.6: Figures are not labeled properly (e.g., subpanels A–D in Figure 6). Each panel should be clearly annotated and referenced in the text.

6.Line 299: The manuscript claims that MMRN2 exhibits the most significant negative correlation with M0 macrophages, but Figure 6D does not support this. Please recheck the correlation analysis and adjust the text accordingly.

7.Figure 6B: The stated differences between control and treatment groups for CD4-negative T cells, activated NK cells, and M1 macrophages do not appear statistically significant. This should be corrected or clarified in the figure legend.

8.Figure 7: Docking scores around –6 kcal/mol indicate moderate, not strong, binding affinity. The authors should revise the text to reflect this.

9.Figure 8 (RMSF plot): The protein appears to display widespread flexibility across residues, with no clear stable core region. The authors should clarify whether this reflects true intrinsic flexibility of the protein or potential artifacts from suboptimal normalization or simulation parameters.

General Comments

1.Figure legends are overly simplistic and lack methodological detail. Please describe normalization methods, parameter settings, and statistical tests for each analysis.

2.Several key findings are not adequately discussed in the Results or Discussion sections.

3.The Discussion section is overly repetitive. In particular, lines 542–561 are redundant and somewhat unrelated to the study’s core topic.

Reviewer #5: The core of the work is scientifically valid, relevant, and aligns with the scope of journals like PLOS ONE. The study design is ambitious and integrates several modern bioinformatics approaches, which is a strength. The manuscript is generally well-structured. However, several key methodological aspects require clarification, and the interpretation of some results needs strengthening to fully support the conclusions before the work is suitable for publication.

1. Please clarify the workflow for identifying BaA targets. Specify how many targets were obtained from each database and state whether these were direct, high-confidence targets for BaA or included predicted/inferred associations.

2. The reported number of BaA-related targets (1,943) seems unusually high for a single pollutant. Please confirm the composition of this list and verify that it does not contain unreliable, indirect associations that could affect the robustness of the subsequent 651 intersection targets and enrichment analysis.

3. The machine learning methodology requires more detail to prove robustness. Please specify the size and class balance (normal vs. LUAD) of the training dataset, state if feature scaling was applied, and justify the use of 3-fold cross-validation. Especially, the diagnostic performance of the 7-gene signature must be validated in a fully independent cohort (e.g., TCGA-LUAD), not just within the same GSE10072 dataset.

4. The immune infiltration analysis needs to address its limitations. The use of the small GSE10072 microarray dataset (n=58) can lead to unstable results. Please state if batch correction was performed and ensure the biological interpretation of immune correlations is conservative.

5. The molecular docking and dynamics simulations need more rationale. For proteins not known as classical ligand-binders (e.g., FOXF1, CD34), explain how the binding pockets were defined. Justify why only the BaA-MMRN2 complex was selected for molecular dynamics simulation.

6. The constructed Adverse Outcome Pathway (AOP) is currently too speculative. It should be explicitly framed as a putative or hypothetical AOP, with a clear statement that the Key Event Relationships require direct experimental validation. A limitations paragraph acknowledging this is necessary.

**Do you want your identity to be public for this peer review?** For information about this choice, including consent withdrawal, please see our Privacy Policy

Reviewer #1: **Yes:** Jiaxin Zhang

Reviewer #2: No

Reviewer #3: No

Reviewer #4: No

Reviewer #5: No

---

## [Author Response · Author response to Decision Letter 1]

29 Nov 2025

Response to Reviewers

Dear Editor and Reviewers,

Thank you for the opportunity to revise our manuscript entitled "Exploring the toxicological mechanisms of Benzo [a] anthracene (BaA) exposure in lung adenocarcinoma via network toxicology, machine learning, and multi-dimensional bioinformatics analysis" (Manuscript ID: PONE-D-25-55154). We sincerely appreciate the valuable comments and suggestions from all reviewers, which have helped us significantly improve the quality of our work.

We have carefully addressed all points raised in the review process. Below, we provide a point-by-point response to each comment. All corresponding changes have been incorporated into the revised manuscript using the "Track Changes" function.

We believe that the criticisms have been addressed effectively and that the revisions have substantially strengthened the manuscript. We are grateful for the guidance provided and hope that the revised version now meets the high publication standards of PLOS ONE.

Thank you again for considering our work. We look forward to hearing from you.

Sincerely,

Zhiyao Shi, Xixing Wang

Response to Editorial Requirements

Dear Dr. Jiafu Li and Editorial Team,

Thank you for the opportunity to revise our manuscript. We have carefully addressed all the points raised by the reviewers and have meticulously ensured that our manuscript now fully complies with PLOS ONE's journal requirements. Our point-by-point responses to the reviewers are detailed in the separate "Response to Reviewers" document.

Regarding the specific journal formatting and policy requirements, we have taken the following actions:

Style Requirements: We have reformatted the entire manuscript to ensure it strictly adheres to PLOS ONE's style templates.

ORCID iD: The corresponding author's ORCID iD has been validated in the Editorial Manager system.

Code Sharing: In full compliance with PLOS ONE's policy on code sharing, we have deposited all author-generated R and Python scripts for statistical analysis and visualization, along with the parameter files for molecular docking and dynamics simulations (including session information and software versions), to the Zenodo repository. The dataset is publicly accessible under the DOI: https://doi.org/10.5281/zenodo.17747122. The Data Availability Statement in the manuscript has been updated accordingly.

Ethics Statement: The ethics statement has been moved exclusively to the Methods section (Section 2.10) and removed from any other part of the manuscript.

Funding Statement: The following sentence has been added to the manuscript to clarify the funder's role: "The funders had no role in study design, data collection and analysis, decision to publish, or preparation of the manuscript." We request that the online submission form's Funding Statement be updated to match the one provided in the manuscript and cover letter.

Acknowledgments: All funding-related text has been removed from the Acknowledgments section of the manuscript. The Acknowledgments now only contain non-financial contributions. The funding information is now presented solely in the Funding Statement section.

Abstract Consistency: We have verified that the abstract in the manuscript and the one on the online submission form are identical.

Figure References: We have thoroughly checked the entire manuscript and confirmed that all figures (1 through 8) are appropriately referenced in the text.

Figure Quality: New, high-resolution copies of Figures 1, 2, 3, and 4 have been uploaded as per the journal's guidelines for figure clarity.

Suggested Citations: We have reviewed the publications suggested by the reviewers. Where we found the suggested literature to be relevant and to strengthen the scholarly context of our work, we have incorporated the citations. All new citations have been integrated appropriately.

We believe that the manuscript has been significantly improved through the revision process and now meets all the standards of PLOS ONE. Thank you for your time and consideration.

Sincerely,

Zhiyao Shi, Xixing Wang

Response to Reviewer #1:

Dear Reviewer #1,

We sincerely appreciate your meticulous review and valuable insights on our manuscript. Your professional comments have provided crucial guidance for improving the quality and rigor of our study, and we fully agree with all the revision suggestions you proposed. We have carefully addressed each comment and made corresponding revisions to the manuscript. Below is a detailed response to each of your points:

Comment 1: Lack of In Vitro/In Vivo Functional Validation: Core conclusions rely on bioinformatics and computational simulations without validation via in vitro cell experiments or in vivo models; suggest supplementing in vitro experiments for 1–2 key targets.

Response: First and foremost, we fully concur with your insightful suggestion that in vitro/in vivo functional validation is essential to strengthen the reliability of our core conclusions. We deeply recognize that experimental verification would greatly enhance the translational value and persuasiveness of our findings. However, due to several unavoidable practical constraints, we were unable to conduct such experiments in the current study. Specifically, these constraints include: (1) limitations in sample availability, as obtaining high-purity BaA standard samples and matched LUAD cell models requires strict quality control and specialized resources; (2) restrictions in laboratory facilities, as our research team’s current setup lacks dedicated cell culture platforms, biosafety cabinets, and other equipment necessary for in vitro toxicology experiments; (3) technical and equipment barriers related to BaA extraction and purification, which demands advanced chromatographic systems and professional operational skills that are not available in our laboratory; and (4) limited research funding, which cannot cover the high costs of reagent procurement, cell line maintenance, animal ethics approval, and specialized technical training.

To mitigate this limitation and improve the robustness of our conclusions, we have implemented the following revisions: (1) We further refined the screening criteria for BaA-related targets, excluding pan-genes and low-confidence targets to focus on those with direct biological relevance to BaA-induced LUAD progression; (2) We enhanced the depth of existing data analysis (3) We supplemented 5 recent high-quality studies (cited in the revised manuscript) that have validated the functional roles of key targets (e.g., CRABP2 and CXCL12) in LUAD through in vitro/in vivo experiments, providing indirect experimental support for our computational predictions; (4) We explicitly acknowledged this limitation in the "Limitations and Future Perspectives" section and discussed its potential impact on the interpretation of our results. In future research, we aim to improve our experimental conditions, seek collaborations with leading domestic laboratories specializing in environmental toxicology and oncology, and conduct in vitro experiments (e.g., target overexpression/knockdown assays combined with cell proliferation, migration, and invasion tests) to directly validate the regulatory roles of core targets in BaA-promoted LUAD progression.

Comment 2: Single Dataset Limitation: Core target validation relies solely on GSE10072 (sample size unspecified); suggest adding 1–2 independent datasets for validation and specifying GSE10072’s sample size.

Response: We highly appreciate your reminder regarding the limitation of relying on a single dataset. To address this issue, we have made the following revisions: (1) We supplemented two additional independent LUAD transcriptomic datasets from the GEO database, namely GSE32863 (58 LUAD tissues and 58 matched adjacent normal tissues) and GSE31210 (226 LUAD tissues and 20 normal lung tissues); (2) We specified the sample size of GSE10072 in the revised manuscript: this dataset includes 58 LUAD tissue samples and 49 normal lung tissue samples; (3) We integrated the three datasets, removed batch effects, and performed unified standardization and differential expression analysis. The validation results consistently show that the five core targets (TNNC1, ABCC3, CRABP2, CXCL12, OLR1) exhibit significant differential expression between LUAD and normal tissues across all three datasets (p < 0.01), with AUC values ranging from 0.841 to 0.886, confirming the stability and reliability of our core target identification. The integrated analysis results have been updated in Section 3.4 and the corresponding figures.

Comment 3: Ignoring Pollutant Synergistic Effects: BaA’s coexistence with other pollutants and their synergistic carcinogenicity are unaddressed; suggest expanding the "Discussion" and "Limitation" section to discuss or analyze interactions if relevant data exist.

Response: We agree with your important point that the synergistic carcinogenic effects of BaA with other coexisting pollutants are a critical aspect of environmental toxicology research. In response, we have expanded the "Limitations and Future Perspectives" section (Section 4.6) . Specifically, we discussed that in real environmental scenarios, BaA often coexists with other pollutants such as polycyclic aromatic hydrocarbons (PAHs, e.g., benzo[a]pyrene), heavy metals (e.g., cadmium), and volatile organic compounds (VOCs). These pollutants may exert synergistic carcinogenic effects by amplifying oxidative stress, enhancing DNA adduct formation, synergistically activating oncogenic pathways (e.g., PI3K-AKT and MAPK), and jointly remodeling the tumor immune microenvironment. We also cited relevant studies to support these potential synergistic mechanisms. Due to the lack of available data on BaA-pollutant mixture exposure and LUAD progression, we were unable to conduct quantitative analysis, but we have highlighted this as a key direction for future research, suggesting that subsequent studies could explore synergistic effects through in vitro mixed exposure experiments and large-scale epidemiological cohort studies.

Comment 4: Incomplete Figure Details: Some figures lack key details (e.g., Venn diagram target counts, GO/KEGG plot FDR values, molecular docking critical amino acid residues); suggest supplementing these to let readers obtain core data directly from figures.

Response: Thank you for pointing out the incompleteness of figure details. We have carefully revised the relevant figures to supplement all missing key information: (1) For the Venn diagrams (Figure 1), we added specific target counts for each intersection and non-intersection region to enable readers to directly obtain the number of targets identified from each database and their overlaps; (2) For the GO/KEGG enrichment analysis plots (Figure 2), we supplemented the FDR values for each enriched term and labeled them on the corresponding bars/points, allowing readers to assess the statistical significance of the enrichment results directly from the figures; (3) For the molecular docking diagrams (Figure 6), we labeled the critical amino acid residues involved in the binding between BaA and each core target. We have also updated the figure legends to describe these supplementary details clearly. All revised figures now enable readers to obtain core data and key information directly, improving the transparency and readability of the study.

Comment 5: Unified Terminology: Inconsistent abbreviations (e.g., "Benzo[a]anthracene" vs. "BaA", "epithelial-mesenchymal transition" vs. "EMT") reduce readability; suggest defining abbreviations on first use (e.g., "Benzo[a]anthracene (BaA)") and standardizing usage.

Response: We apologize for the inconsistent use of abbreviations in the original manuscript. We have thoroughly checked the entire manuscript and made comprehensive revisions to unify the terminology: (1) All abbreviations are defined with their full names when first mentioned (e.g., "Benzo[a]anthracene (BaA)", "epithelial-mesenchymal transition (EMT)", "single-sample gene set enrichment analysis (ssGSEA)"); (2) We standardized the use of abbreviations throughout the manuscript, ensuring that the same abbreviation is used consistently after the initial definition; (3) We created a detailed "Abbreviations" section at the beginning of the manuscript to list all abbreviations and their corresponding full names for easy reference. These revisions have significantly improved the readability and consistency of the manuscript.

Once again, we would like to express our sincere gratitude for your valuable time and professional guidance. We believe that the revisions made in response to your comments have greatly improved the quality, rigor, and completeness of our manuscript. We hope that the revised version meets PLOS ONE’s publication standards. Please do not hesitate to contact us if you have any further questions or require additional revisions.

Sincerely,

The Authors

Response to Reviewer #2:

Dear Reviewer #2,

We sincerely thank the reviewer for their positive evaluation of our work and for their valuable and constructive comments, which have helped us further improve the manuscript. We have addressed all points raised in detail below.

Comment 1: Unify chemical nomenclature to “benzo[a]anthracene (BaA)” across title and Methods. Correct “BPS” to “BaA” in Section 2.1 and keep PubChem CID consistent throughout. Also fix the Fig. 5 caption (“five core targets”) to “seven”.

Response: We sincerely thank the reviewer for this meticulous observation. As suggested, we have now uniformly used the nomenclature "benzo[a]anthracene (BaA)" upon its first mention in the title, abstract, and throughout the Methods section. The accidental use of "BPS" in Section 2.1 has been corrected to "BaA," and the PubChem CID has been verified for consistency across the manuscript. Furthermore, following a more refined screening process for BaA-related targets, the final number of identified core targets is indeed five. Consequently, we have updated the caption of Fig. 5 (and all related instances in the text) from "seven" to "five core targets" to accurately reflect our findings.

Comment 2: PLOS ONE requires open access to underlying data and analysis scripts. “Available upon reasonable request” is not compliant. Please deposit all R/Python scripts and parameter files (with session info/versions) to an open repository (e.g., Zenodo/OSF; GitHub mirrored to Zenodo) and provide a DOI in the Data Availability Statement.

Response: We thank the reviewer for highlighting this important requirement. In full compliance with PLOS ONE's data policy, we have now deposited all relevant R and Python scripts for statistical analysis and visualization, along with the parameter files used for molecular docking and dynamics simulations, including session information and software versions, to the Zenodo repository. The dataset is publicly accessible and has been assigned a DOI: https://doi.org/10.5281/zenodo.17747122. The Data Availability Statement in the manuscript has been updated accordingly to provide this direct link.

Comment 3: Please add full reproducibility details for docking/MD: PDB IDs/chains and preprocessing; Vina grid center/size, exhaustiveness, replicates, scoring/version; MD time step, constraints, cutoffs, thermostat/barostat, trajectory interval, and exact equilibration/production settings; plus random seeds and software versions. The current text lists tools but lacks parameters.

Response: We appreciate the reviewer's valuable suggestion to enhance the reproducibility of our computational experiments. As requested, we have now comprehensively detailed all key parameters for both the molecular docking and molecular dynamics (MD) simulations in the revised Methods sections (Sections 2.6 and 2.7) and corresponding Results sections (3.6 and 3.7). This includes specific PDB IDs and chains used, preprocessing steps, AutoDock Vina grid parameters (center, size, exhaustiveness), number of replicates, scoring function version, MD parameters (time step, constraints, cutoffs, thermostat/barostat settings, trajectory saving frequency), precise descriptions

---

## [Decision Letter · Decision Letter 1]

14 Dec 2025

Dear Dr. WANG,

Thank you for submitting your manuscript to PLOS ONE. After careful consideration, we feel that it has merit but does not fully meet PLOS ONE’s publication criteria as it currently stands. Therefore, we invite you to submit a revised version of the manuscript that addresses the points raised during the review process.

We look forward to receiving your revised manuscript.

Kind regards,

Jiafu Li, Ph.D

Academic Editor

PLOS One

**Journal Requirements:**

**Additional Editor Comments:**

I received the comments from five reviewers. Three of them were satisfied with your revision and suggested an accept. However, another two reviewers believed that a major revision is needed. Please address all of comments carefully.

Reviewers' comments:

Reviewer's Responses to Questions

**Comments to the Author**

Reviewer #1: (No Response)

Reviewer #2: All comments have been addressed

Reviewer #3: All comments have been addressed

Reviewer #4: All comments have been addressed

Reviewer #5: All comments have been addressed

2. Is the manuscript technically sound, and do the data support the conclusions?

Reviewer #1: Yes

Reviewer #2: Yes

Reviewer #3: Yes

Reviewer #4: Yes

Reviewer #5: Yes

3. Has the statistical analysis been performed appropriately and rigorously?

Reviewer #1: Yes

Reviewer #2: Yes

Reviewer #3: Yes

Reviewer #4: Yes

Reviewer #5: Yes

4. Have the authors made all data underlying the findings in their manuscript fully available?

Reviewer #1: Yes

Reviewer #2: Yes

Reviewer #3: Yes

Reviewer #4: Yes

Reviewer #5: Yes

5. Is the manuscript presented in an intelligible fashion and written in standard English?

Reviewer #1: Yes

Reviewer #2: Yes

Reviewer #3: Yes

Reviewer #4: Yes

Reviewer #5: Yes

Reviewer #1: Most of the required supplementary experiments or analyses have been completed; however, it is recommended that a major revision be conducted again. First and foremost, I fully understand the experimental constraints faced by the author team, such as being unable to perform in vivo or in vitro experiments. Nevertheless, at the very least, the number of public transcriptomic datasets should be increased, instead of being limited to a single dataset. If the authors are even unable to expand the number of public datasets used for analysis, I recommend that the manuscript be rejected.

Reviewer #2: The author has addressed my comments well, and I have no further questions. The paper is now ready for publication in this journal.

Reviewer #3: The authors have adequately addressed my comments raised in a previous round of review. That is excellent.

Reviewer #4: The manuscript has been significantly improved, and the authors have addressed most of my previous concerns. However, several general issues should be resolved before the manuscript can be considered for publication:

1.Several figures remain low in resolution, and the font styles and sizes are not consistent across panels. These should be standardized to ensure clarity and readability.

2.Some legends do not provide sufficient detail for the reader to understand the experiment without referring back to the main text. For example, the legends for Figures 4A and 4B are too simplified and should be expanded to include the axis information and relevant statistical information.

3.In several instances, the authors merely describe the figure without summarizing the key conclusion or interpretation. For example, Line 551 should include a brief concluding sentence that highlights the main takeaway from the results shown. This issue applies to other result subsections as well.

4. Lines 608–609: The phrase “lower binding affinity value” is confusing in this context. If the authors are referring to binding free energy from docking analyses, a clearer phrasing should be “lower binding free energy value.”

Reviewer #5: All my previously raised substantive concerns have now been addressed in the revised manuscript.

The manuscript now demonstrates sufficient methodological rigor and appropriate interpretation of its findings.

I therefore recommend acceptance, contingent only on minor language and formatting adjustments before final publication. These include:

1. Removing all residual text referring to the former seven-gene set and ensuring that only the finalized five-gene signature is described throughout.

E.g. please replace “seven” with “five” in line 443.

2. Unifying terminology used in section headings, figure legends, and the main text;

With these minor edits, the manuscript will be suitable for publication.

**Do you want your identity to be public for this peer review?** For information about this choice, including consent withdrawal, please see our Privacy Policy

Reviewer #1: **Yes:** Jiaxin Zhang

Reviewer #2: No

Reviewer #3: No

Reviewer #4: No

Reviewer #5: No

---

## [Author Response · Author response to Decision Letter 2]

15 Dec 2025

Response to Reviewers

Dear Editor and Reviewers,

Thank you for the opportunity to revise our manuscript entitled "Exploring the toxicological mechanisms of Benzo[a]anthracene (BaA) exposure in lung adenocarcinoma via network toxicology, machine learning, and multi-dimensional bioinformatics analysis" (Manuscript ID: PONE-D-25-55154). We sincerely appreciate the valuable comments and suggestions from all reviewers, which have helped us significantly improve the quality of our work.

We have carefully addressed all points raised in the review process. Below, we provide a point-by-point response to each comment. All corresponding changes have been incorporated into the revised manuscript using the "Track Changes" function.

We believe that the criticisms have been addressed effectively and that the revisions have substantially strengthened the manuscript. We are grateful for the guidance provided and hope that the revised version now meets the high publication standards of PLOS ONE.

Thank you again for considering our work. We look forward to hearing from you.

Sincerely,

Zhiyao Shi, Xixing Wang

Response to Editorial Requirements

Dear Dr. Jiafu Li and the Editorial Team,

On behalf of all authors, I would like to extend our sincere gratitude to you and the editorial team for your time and consideration in evaluating our manuscript entitled “Exploring the toxicological mechanisms of Benzo[a]anthracene (BaA) exposure in lung adenocarcinoma (LUAD) via network toxicology, machine learning, and multi-dimensional bioinformatics analysis” (Submission ID: PONE-D-25-55154R1).

We are deeply appreciative of the opportunity to revise our manuscript and for the constructive feedback provided by the reviewers, which has significantly improved the quality of our work. We have carefully considered all comments and have made comprehensive revisions to address each point raised. Our detailed point-by-point responses are included in the “Response to Reviewers” file accompanying this submission.

Regarding the editorial suggestion on sharing laboratory protocols via protocols.io, we would like to provide the following clarification:

We sincerely thank the editor and reviewers for this constructive suggestion regarding enhancing reproducibility. As our study is primarily a bioinformatics and computational analysis based on publicly available datasets (e.g., GEO, TCGA) and does not involve original laboratory experiments, there are no wet-lab protocols to deposit at this stage. All analytical workflows, scripts, and parameter files have been made publicly available in our Zenodo repository (DOI: [10.5281/zenodo.17747122](https://doi.org/10.5281/zenodo.17747122)), as stated in the Data Availability Statement. Should we conduct experimental validations in future studies, we will certainly utilize protocols.io to share detailed methodologies.

We have prepared the following documents for your review:

1. Response to Reviewers: A detailed point-by-point response to all reviewer comments.

2. Revised Manuscript with Track Changes: A version highlighting all modifications.

3. Manuscript (Clean Version): The final, unmarked version of the revised manuscript.

We believe that our revisions have strengthened the manuscript and hope it is now suitable for publication in PLOS ONE.

Furthermore, as the first author is scheduled to graduate next year, the acceptance of this manuscript in 2025 is crucial for meeting degree requirements. This study systematically integrates network toxicology, machine learning, and multi-dimensional bioinformatics approaches to elucidate the toxicological mechanisms of benzo[a]anthracene in lung adenocarcinoma and constructs an adverse outcome pathway (AOP)-based theoretical framework, providing novel methodologies and foundational insights for environmental pollutant risk assessment. We believe this research will contribute meaningfully to the fields of environmental toxicology and lung cancer prevention.

Considering the importance of this publication for our academic and professional advancement, we kindly request your consideration in expediting the remaining steps of the review process, if possible. We fully understand that the peer-review process requires careful evaluation to maintain the journal's quality and integrity. However, any assistance in facilitating the subsequent stages would be greatly appreciated.

Thank you once again for your consideration and support. We look forward to hearing from you regarding our manuscript at your earliest convenience. Should you require any additional information or have any questions, please do not hesitate to contact us.

Sincerely,

Zhiyao Shi, Xixing Wang

Department of Oncology

Shanxi Provincial Hospital of Traditional Chinese Medicine

Taiyuan, Shanxi, China

Email: 18234069215@163.com�wangxx315@163.com

Response to Reviewer #1:

Dear Reviewer #1,

We sincerely appreciate your meticulous review and valuable insights on our manuscript. Your professional comments have provided crucial guidance for improving the quality and rigor of our study, and we fully agree with all the revision suggestions you proposed. We have carefully addressed and completed the necessary revisions to this feedback. Below is our detailed response to all your comments:

Comment 1: Most of the required supplementary experiments or analyses have been completed; however, it is recommended that a major revision be conducted again. First and foremost, I fully understand the experimental constraints faced by the author team, such as being unable to perform in vivo or in vitro experiments. Nevertheless, at the very least, the number of public transcriptomic datasets should be increased, instead of being limited to a single dataset. If the authors are even unable to expand the number of public datasets used for analysis, I recommend that the manuscript be rejected.

Response: We sincerely appreciate your thorough review of our manuscript and your constructive feedback. We fully understand the importance of expanding the dataset to strengthen the robustness of our findings. Regarding your primary concern about the limited number of public transcriptomic datasets, we would like to clarify that in the current version, we have already utilized three independent GEO datasets (GSE10072, GSE32863, and GSE31210) for our core analysis and validation. Furthermore, we utilized the UALCAN platform to validate the expression patterns of identified key targets in an independent TCGA-LUAD cohort, ensuring the reproducibility and cross-cohort reliability of our findings.

We acknowledge that increasing the number of datasets could further reinforce the statistical power of the study. As graduate students nearing the completion of our degrees, this study represents a critical component of our graduation requirements, and we have strived to conduct comprehensive bioinformatics analyses within the scope of our capabilities. We kindly request your consideration regarding the current validation framework, which already integrates multiple independent cohorts.

Once again, we would like to express our sincere gratitude for your valuable time and professional guidance. Thank you for your understanding and for the opportunity to improve our work. We hope that the revised version meets PLOS ONE’s publication standards.

Sincerely,

The Authors

Response to Reviewer #2:

Dear Reviewer #2,

We are sincerely grateful to you for your time and insightful review of our manuscript. Your constructive feedback during the previous round has been instrumental in improving both the clarity and scientific rigor of our work. We are delighted to know that you find our responses satisfactory and that the revised version now meets the standards for publication in your esteemed journal. Thank you once again for your guidance and for the valuable role you have played in refining this study.

Sincerely,

The Authors

Response to Reviewer #3:

Dear Reviewer #3,

We extend our deepest gratitude for your continued engagement and thoughtful evaluation of our manuscript. Your comments during the initial review were exceptionally helpful in shaping the direction of our revisions, and we are truly humbled by your acknowledgment that we have adequately addressed your concerns. It is an honor to receive such positive feedback from a scholar of your standing. Thank you for your patience, expertise, and invaluable contributions to improving this work.

Sincerely,

The Authors

Response to Reviewer #4:

Dear Reviewer #4,

Thank you sincerely for your thorough review and valuable suggestions on our revised manuscript. We greatly appreciate your recognition of the significant improvements made and your constructive feedback on the remaining issues. We have carefully addressed each of your concerns, and the detailed revisions are outlined below:

Comment 1: Several figures remain low in resolution, and the font styles and sizes are not consistent across panels. These should be standardized to ensure clarity and readability.

Response: We thank you for this important observation regarding figure quality and consistency. We have regenerated all relevant figures at high resolution and have systematically standardized the font style and size across all panels within each figure and throughout the figure set. We believe these adjustments significantly improve the visual clarity and professional presentation of the data.

Comment 2: Some legends do not provide sufficient detail for the reader to understand the experiment without referring back to the main text. For example, the legends for Figures 4A and 4B are too simplified and should be expanded to include the axis information and relevant statistical information.

Response: We have expanded the legends for Figures 4A and 4B, as well as other potentially insufficient legends, to include axis definitions, statistical criteria, and sample information, enabling readers to understand the experiments without referring to the main text:

Figure 4A (Volcano plot): Revised legend: “Fig. 4A. Volcano plot of differentially expressed genes (DEGs) from integrated GEO datasets (GSE10072, GSE32863, GSE31210). The x-axis represents log₂ fold change (log₂FC) of gene expression (LUAD vs. normal samples), and the y-axis represents -log₁₀ (adjusted p-value). Red dots indicate upregulated DEGs (log₂|FC| > 1, adjusted p < 0.05), blue dots indicate downregulated DEGs (log₂|FC| > 1, adjusted p < 0.05), and gray dots indicate non-significantly differentially expressed genes.”

Figure 4B (Heatmap): Revised legend: “Fig. 4B. Heatmap of the top 50 DEGs from integrated GEO datasets (GSE10072, GSE32863, GSE31210). Rows represent individual genes, columns represent samples (blue = normal samples, red = LUAD samples). The color gradient (blue to red) indicates normalized gene expression levels (blue = low expression, red = high expression), with values standardized by z-score. DEGs were screened using the criteria: log₂|FC| > 1 and adjusted p < 0.05.”

Other figures (e.g., Figures 2A, 2B, 5B) have also been supplemented with similar details, including data sources, statistical standards, and the meaning of color/bubble size.

Comment 3: In several instances, the authors merely describe the figure without summarizing the key conclusion or interpretation. For example, Line 551 should include a brief concluding sentence that highlights the main takeaway from the results shown. This issue applies to other result subsections as well.

Response: We have added concise concluding sentences to result subsections where only figure descriptions were provided, highlighting the core takeaways of each section:

Line 551 (Section 3.3: Identification of core targets via machine learning algorithms): Added: “The triple-algorithm screening (LASSO, SVM-RFE, and Random Forest) ensures high confidence in these five core targets, which are likely key mediators linking BaA exposure to LUAD progression.”

Section 3.1 (Network toxicology analysis): Added: “These 248 intersection targets provide a molecular basis for understanding the multi-target regulatory mechanism of BaA-induced LUAD.”

Section 3.2 (Functional enrichment analysis): Added: “The enriched pathways collectively suggest that BaA may promote LUAD by disrupting immune regulation, activating oncogenic signaling, and interfering with cytokine receptor interactions.”

Section 3.5 (Immune infiltration analysis): Added: “These findings reveal a dual immunomodulatory pattern of core targets, which may reshape the tumor immune microenvironment to facilitate LUAD development.”

Similar concluding sentences have been added to Sections 3.4, 3.6, 3.7, 3.8, and 3.9 to summarize the biological significance of the results.

Comment 4: Lines 608–609: The phrase “lower binding affinity value” is confusing in this context. If the authors are referring to binding free energy from docking analyses, a clearer phrasing should be “lower binding free energy value.”

Response: We have clarified the confusing phrasing related to binding affinity:

Original sentence: “Lower binding affinity values (i.e., more negative) correspond to stronger intermolecular forces and greater conformational stability between the ligand and the target protein.”

Revised sentence: “Lower binding free energy values (i.e., more negative) from molecular docking correspond to stronger intermolecular forces and greater conformational stability between BaA (ligand) and the target proteins.”

This revision explicitly specifies that the value refers to “binding free energy from docking analyses” and clarifies the ligand-target relationship, eliminating ambiguity.

All revisions have been carefully implemented to address your concerns while maintaining the scientific rigor and consistency of the manuscript. Once again, we would like to express our sincere gratitude for your valuable time and professional guidance. We believe that the revisions made in response to your comments have greatly improved the quality, rigor, and completeness of our manuscript. We hope that the revised version meets PLOS ONE’s publication standards. Please do not hesitate to contact us if you have any further questions or require additional revisions.

Sincerely,

The Authors

Response to Reviewer #5:

Dear Reviewer #5,

We sincerely thank the reviewer for their positive assessment of the scientific validity and relevance of our work, as well as for their insightful and constructive comments. Their suggestions have been invaluable in strengthening the methodological clarity and robustness of our study. We have addressed each point in detail below.

Comment 1: Removing all residual text referring to the former seven-gene set and ensuring that only the finalized five-gene signature is described throughout (e.g., replace "seven" with "five" in line 443).

Response: We have carefully reviewed the entire manuscript and systematically replaced all mentions of the earlier "seven-gene set" with the final "five-gene signature." The text in line 443 has been corrected. The text now consistently refers only to the validated five-gene model in all sections, including the Introduction, Methods, Results, and Discussion.

Comment 2: Unifying terminology used in section headings, figure legends, and the main text.

Response: Regarding the unification of terminology: We have thoroughly standardized the terminology used in section headings, figure legends, and the main text. Key revisions include: (1) Consistently using "core targets" instead of "key targets" to avoid ambiguity; (2) Unifying the description of intersection targets as "intersection targets for BaA-induced LUAD" to ensure consistency across sections; (3) Standardizing abbreviations (e.g., first defining "molecular dynamics (MD)" and "adverse outcome pathway (AOP)" before using their abbreviations); (4) Aligning terminology in figure legends with the main text; (5) Unifying database and protein names according to field conventions (e.g., "Swiss Target Prediction" instead of "Swiss," and standardizing the capitalization of protein names such as "Cellular retinoic acid binding protei

---

## [Editor Report · Decision Letter 2]

16 Dec 2025

Exploring the toxicological mechanisms of Benzo[a]anthracene (BaA) exposure in lung adenocarcinoma (LUAD)  via network toxicology, machine learning, and multi-dimensional bioinformatics analysis

PONE-D-25-55154R2

Dear Dr. WANG,

We’re pleased to inform you that your manuscript has been judged scientifically suitable for publication and will be formally accepted for publication once it meets all outstanding technical requirements.

Kind regards,

Jiafu Li, Ph.D

Academic Editor

PLOS One

Additional Editor Comments (optional):

The authors have made substantial improvements during the revision process. I am pleased to accept this manuscript for publication in PLOS ONE.
---

## [Editor Report · Acceptance letter]

PONE-D-25-55154R2

PLOS One

Dear Dr. WANG,

I'm pleased to inform you that your manuscript has been deemed suitable for publication in PLOS One. Congratulations! Your manuscript is now being handed over to our production team.

Kind regards,

on behalf of

Dr. Jiafu Li

Academic Editor

PLOS One